# Vittrup Man–The life-history of a genetic foreigner in Neolithic Denmark

**Anders Fischer**[1,2,3]*, **Karl-Göran Sjögren**[1]*, **Theis Zetner Trolle Jensen**[3], **Marie Louise Jørkov**[4], **Per Lysdahl**[5], **Tharsika Vimala**[3], **Alba Refoyo-Martínez**[3], **Gabriele Scorrano**[3], **T. Douglas Price**[1,6], **Darren R. Gröcke**[7], **Anne Birgitte Gotfredsen**[3], **Lasse Sørensen**[8], **Verner Alexandersen**[4], **Sidsel Wåhlin**[5], **Jesper Stenderup**[3], **Ole Bennike**[9], **Andrés Ingason**[3,10], **Rune Iversen**[11], **Martin Sikora**[3], **Fernando Racimo**[3], **Eske Willerslev**[3,12], **Morten E. Allentoft**[3,13], **Kristian Kristiansen**[1,3]

1 Department of Historical Studies, University of Gothenburg, Gothenburg, Sweden, 2 Sealand Archaeology, Kalundborg, Denmark, 3 Globe Institute, University of Copenhagen, Copenhagen K, Denmark, 4 Laboratory of Biological Anthropology, University of Copenhagen, Copenhagen, Denmark, 5 Vendsyssel Historical Museum, Hjørring, Denmark, 6 Laboratory for Archaeological Chemistry, University of Wisconsin-Madison, Madison, Wisconsin, United States of America, 7 Department of Earth Sciences, Durham University, Durham, England, United Kingdom, 8 Danish National Museum, Copenhagen, Denmark, 9 Geological Survey of Denmark and Greenland, Copenhagen, Denmark, 10 Institute of Biological Psychiatry, Mental Health Services, Copenhagen University Hospital, Copenhagen, Denmark, 11 The Saxo Institute—Section of Archaeology, University of Copenhagen, Copenhagen S, Denmark, 12 Department of Zoology, University of Cambridge, Cambridge, United Kingdom, 13 Trace and Environmental DNA (TrEnD) Laboratory, School of Molecular and Life Sciences, Curtin University, Perth, Australia

* andersfischerkalundborg@hotmail.com (AF); kg.sjogren@archaeology.gu.se (KGS)

**Data Availability Statement:** All relevant data are within the manuscript and its Supporting Information files. Additionally, the following data underlying the results are available: 1) Protein data are available via https://www.proteomexchange.

## Abstract

The lethally maltreated body of Vittrup Man was deposited in a Danish bog, probably as part of a ritualised sacrifice. It happened between c. 3300 and 3100 cal years BC, i.e., during the period of the local farming-based Funnel Beaker Culture. In terms of skull morphological features, he differs from the majority of the contemporaneous farmers found in Denmark, and associates with hunter-gatherers, who inhabited Scandinavia during the previous millennia. His skeletal remains were selected for transdisciplinary analysis to reveal his life-history in terms of a population historical perspective. We report the combined results of an integrated set of genetic, isotopic, physical anthropological and archaeological analytical approaches. Strontium signature suggests a foreign birthplace that could be in Norway or Sweden. In addition, enamel oxygen isotope values indicate that as a child he lived in a colder climate, i.e., to the north of the regions inhabited by farmers. Genomic data in fact demonstrates that he is closely related to Mesolithic humans known from Norway and Sweden. Moreover, dietary stable isotope analyses on enamel and bone collagen demonstrate a fisher-hunter way of life in his childhood and a diet typical of farmers later on. Such a variable life-history is also reflected by proteomic analysis of hardened organic deposits on his teeth, indicating the consumption of forager food (seal, whale and marine fish) as well as farmer food (sheep/goat). From a dietary isotopic transect of one of his teeth it is shown that his transfer between societies of foragers and farmers took place near to the end of his teenage years.

org, identifier PXD044743; 2) a 3D model of a tooth is made available at https://doi.org/10.5281/zenodo.7802122.

**Funding:** MEA: Marie Curie Actions of the European Union FP7/2007-2013, grant no. 300554; https://marie-sklodowska-curie-actions.ec.europa.eu. KK: Swedish Riksbankens Jubileumsfond, grant no. M16-0455:1; https://www.rj.se/en. KK: COREX ERC Synergy grant ID 951385; https://www.corex-erc.com. FR: Villum Fonden Young Investigator award project no. 00025300;https://veluxfoundations.dk/en/16-new-villum-young-investigators-in-2022. FR: Novo Nordisk Fonden Data Science Ascending Investigator Award NNF22OC0076816;Novo Nordisk Fonden Data Science Ascending Investigator Award. The funders had no role in study design, data collection and analysis, decision to publish, or preparation of the manuscript. All necessary permits were obtained for the described study, which complied with all relevant regulations.

**Competing interests:** The authors have declared that no competing interests exist.

## Introduction

Vittrup Man (Fig 1) is the popular archaeological name for the severely fragmented Neolithic skeleton of a 30–40 years old male, who was found in a peat bog in Northwest Denmark (Fig 2). His remains were recovered in 1915 during peat cutting together with a wooden club, a ceramic vessel and bovine bones (SI.1 in S1 File). In 2014 he was incorporated into a large genomic project investigating the Mesolithic and Neolithic gene pools of Eurasia [1]. The realisation that this individual had a genetic profile different from his local contemporaries led to the current study. We present and combine data on genetic ancestry, physical appearance, geographic origin, dietary history and cultural environment. New genetic evidence relating to the Vittrup individual is gained through merging and re-analysis of data that were published separately (Coutinho et al. 2020 [2]; Allentoft et al. 2022 [1]). By using a combination of bioarchaeological methods and traditional archaeological approaches we gain insight into a European Stone Age individual's life-history at unprecedented resolution. The dramatic changes in geography and cultural environments revealed, adds decisive new dimensions to a generations-long discussion on population history, migration and interaction in Scandinavia during the local Middle Neolithic [3–14]. Our results involve further attention to a millennial-long relationship in European prehistory between farmers of southern origin and hunter-fisher-gatherers of northern genetic ancestry.

## Material–The human skeletal remains

### Physical characteristics

The human remains found in the bog of Vittrup include a right ankle bone (*talus*), the shaft of a lower left shin bone (*tibia*), a fragmented skull and a jawbone (*mandibula*) (Figs 1, 3 and 4; SI.1 in S1 File). All of the remains are thought to be from one and the same individual. This is supported by multiple radiocarbon dates and stable isotope analyses conducted on the different skeletal elements as well as the preservation, general appearance and dimensions of the bones. We, therefore, present the remains as a single individual: Vittrup Man.

Based on the cranial sexual dimorphic features, the skull and lower jawbone are determined to be of a male. These features include a sloping forehead, rounded margins of the eye sockets and a pronounced lower forehead (*glabella*). The mastoid process behind the ear opening is large and drop-shaped, and the muscle attachment sites on the temporal bone, as well as the occipital, are very pronounced. The jawbone also displays male characteristics including square chins, angled lateral ends of the lower jawbone (*ramus*) and marked muscle attachments. The osteological sex is confirmed by genetic analysis.

The bone of the skull is extremely dense and measures as much as 10.1 mm in thickness. However, the morphology of the inner and outer table and the middle layer (*diplöe*) appears normal. Similar bone dimensions and density characterises Late Mesolithic skulls from Denmark, Norway and Sweden [15–18], and seem also to be represented in skeletons associated with the Pitted Ware Culture [19 Grab 31]. In contrast, Funnel Beaker Culture associated humans, coeval to Vittrup Man, have significantly thinner skulls.

### Age and health

The size and features of the skull and lower leg bone suggest that this is an adult individual. The 16 teeth available for our study all represent permanent dentition. They display significant wear of enamel and dentine on especially premolars and first molars (SI.2 in S1 File). The enamel is flaky on several teeth in the lower jaw. Still, it is evident that a minimum of one tooth in the upper jaw and four teeth in the lower jaw (Fig 4) have moderate calculus deposits

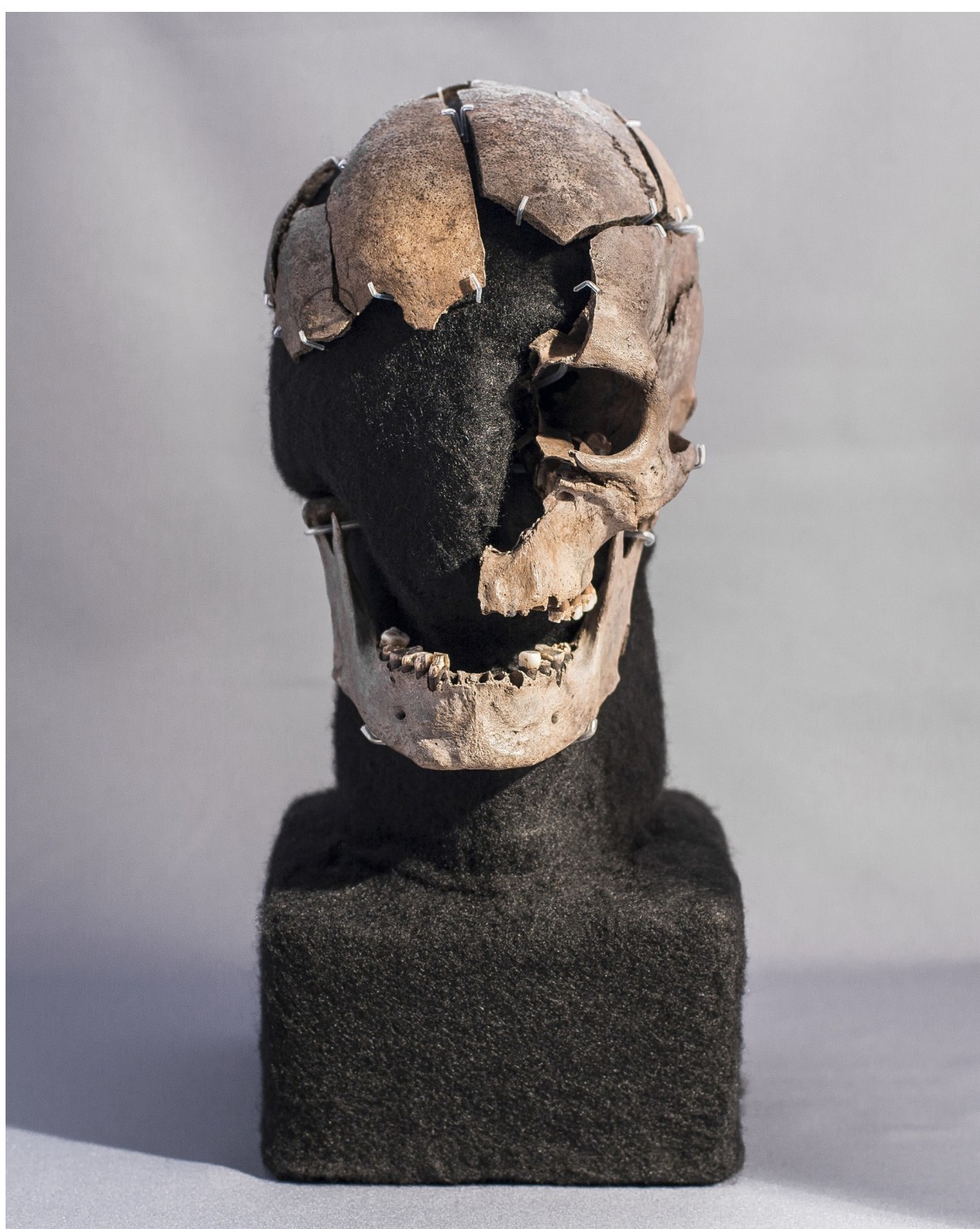

**Fig 1. The cranial remains of Vittrup Man, who ended up in a bog after his skull had been crushed by at least eight heavy blows.** Photo: Stephen Freiheit.

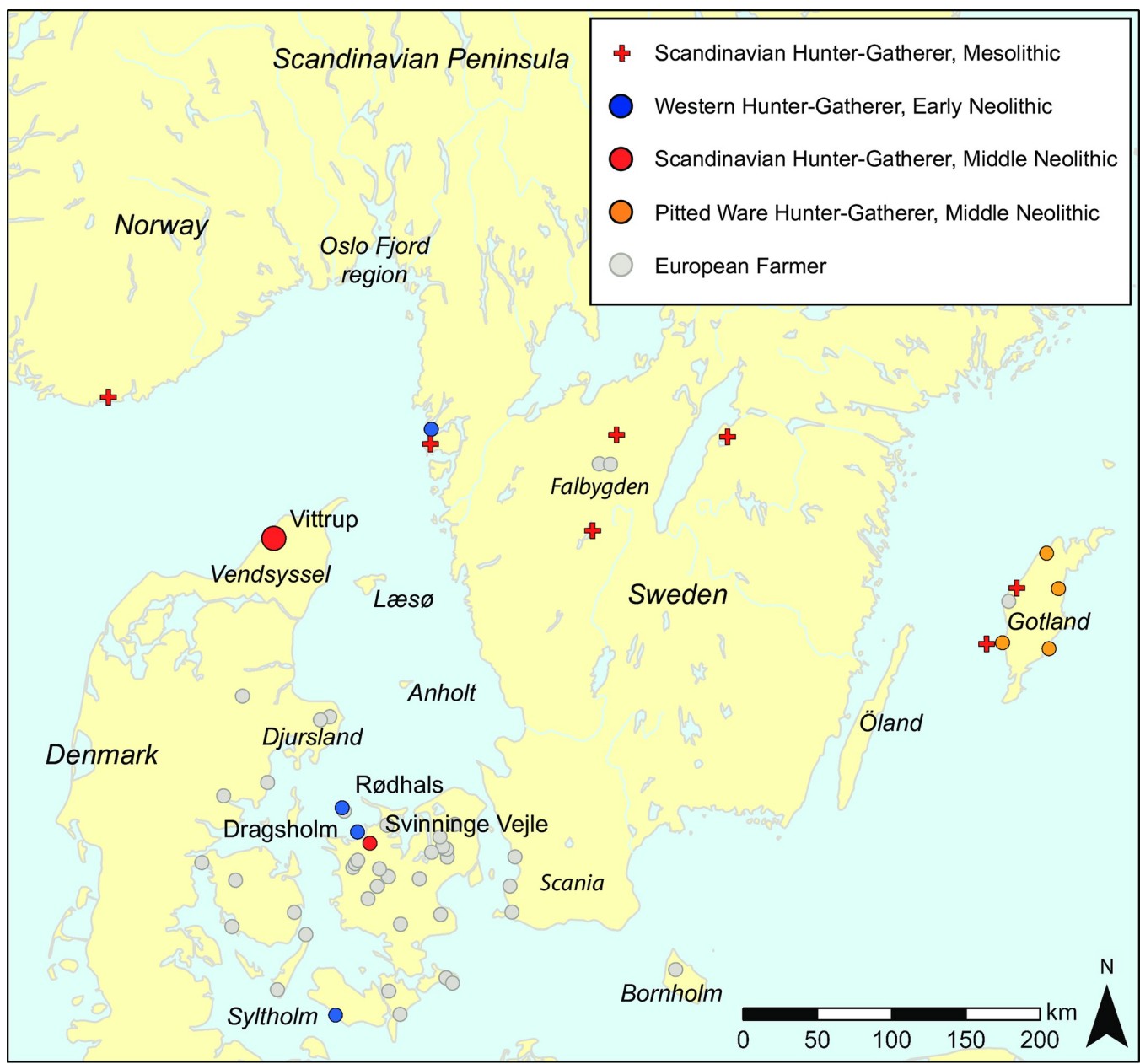

**Fig 2. Southern Scandinavia with the location of humans and areas mentioned in the text.** Dots mark individuals with published genome-wide data and a [14]C date referring to the epoch of the Funnel Beaker Culture, c. 3900–2800 cal. BC. Data from [1, 2]. Map data from www.naturalearthdata.com.

(Fig 1; SI.2 and SI.3 in S1 File). Based on our visual inspections of the remaining dentition no caries lesions can be reported.

Due to the lack of other skeletal elements, age at death can only be assessed from cranial suture closure and tooth wear. Large individual variation applies to the former approach. Stressing this methodological ambiguity, we judge 30–40 years to be a reasonable estimate and consider the low end of this range the most likely [20, 21]. Age determination based on attrition and chipping of teeth is complicated since it is strongly correlated to the dietary regime (SI.2 in S1 File)—which, according to our isotopic analyses, changed significantly for this individual at a time when most of his permanent dentition was already fully developed.

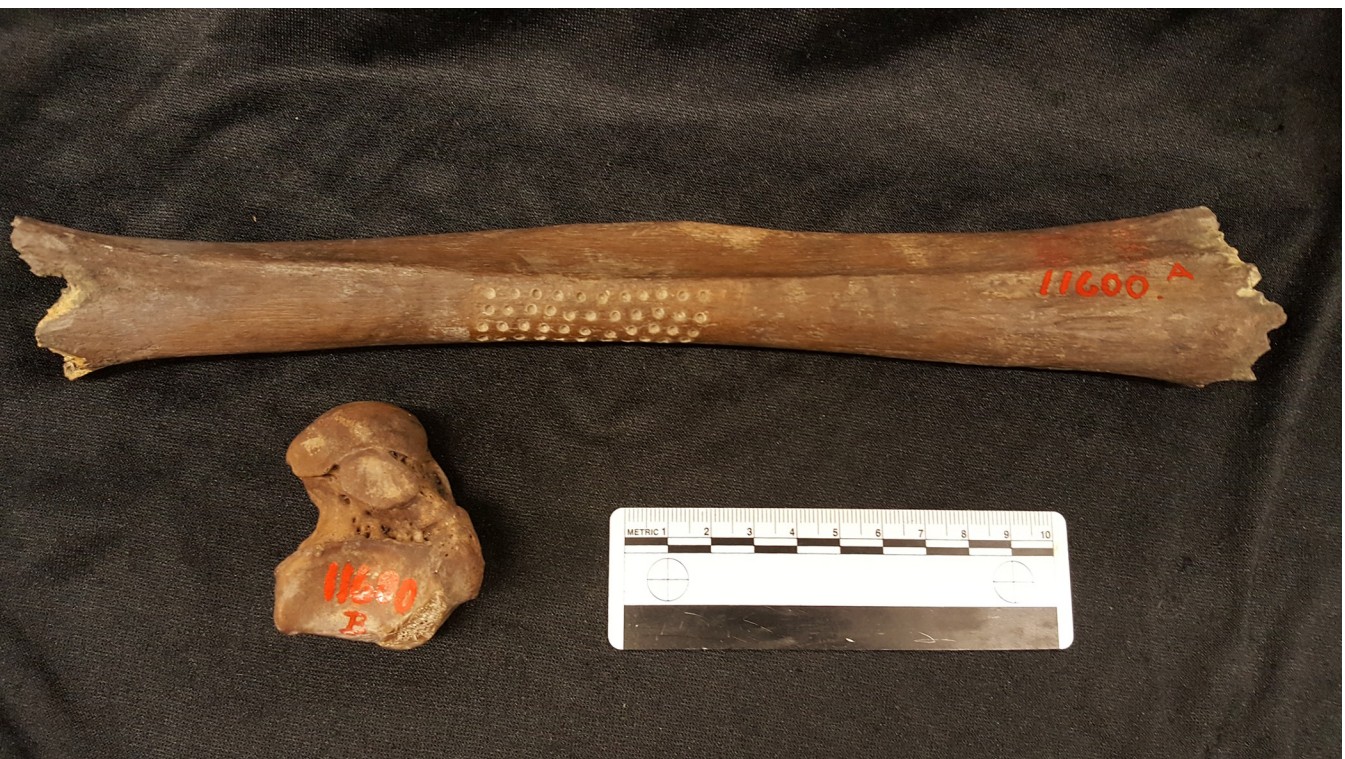

**Fig 3. Left lower leg bone and right ankle bone.** Like the cranial fragments and the lower jawbone these two bones are coloured brown from millennial deposition in the Vittrup mire. Arrows mark where material was taken for $^{14}$C analysis and dietary isotope measurements. Photo: Marie Louise Jørkov.

The outer surface of the cranial vault, more specifically on the top of the head, on the forehead and on the back of the head displays small diffused porosities with rounded margins, characteristic of *porotic hyperostosis*. This condition is nonspecific but has been associated with anaemia and infection [22]. The rounded appearance of the porosity margins suggests that the condition wasn't active at the time of death. Bone porosity in the lower jawbone can also be observed at the attachment sites for the chin muscles and sporadically next to the dentition (Fig 4 and SI.2.1 in S1 File).

## Violent death

The fragmented state of the cranium (Fig 1) is the result of at least eight blows, which split it into several parts (Fig 5). There are no signs of healing–the traumas were obviously fatal.

No marks on the bones from cutting up the dead body are observed. We, therefore, understand the lack of trunk and limb bones to be the result of (post-)depositional circumstances. Fragments of the mistreated body may from the start have been deposited in a dispersed manner, or upon decomposition may have floated apart. Incomplete recovery of bones actually present in the peat dig should also be considered (SI.1 in S1 File).

The impact lesions on the skull are characterised by oval fractures (Fig 6) with larger radiating fracture lines (Figs 1 and 5). Such damages are described in the forensic anthropological literature on skeletal traces of fatal violence—e.g., [23] - and are known from other Neolithic human skeletons [24–26]. They suggest blunt force caused by contact with an object made of more resilient material than the skull bone, possibly with a rounded surface and a diameter of a minimum of 2 cm. The wooden club found next to the skeletal remains (SI.1 in S1 File) would have been a weapon likely to produce these fractures.

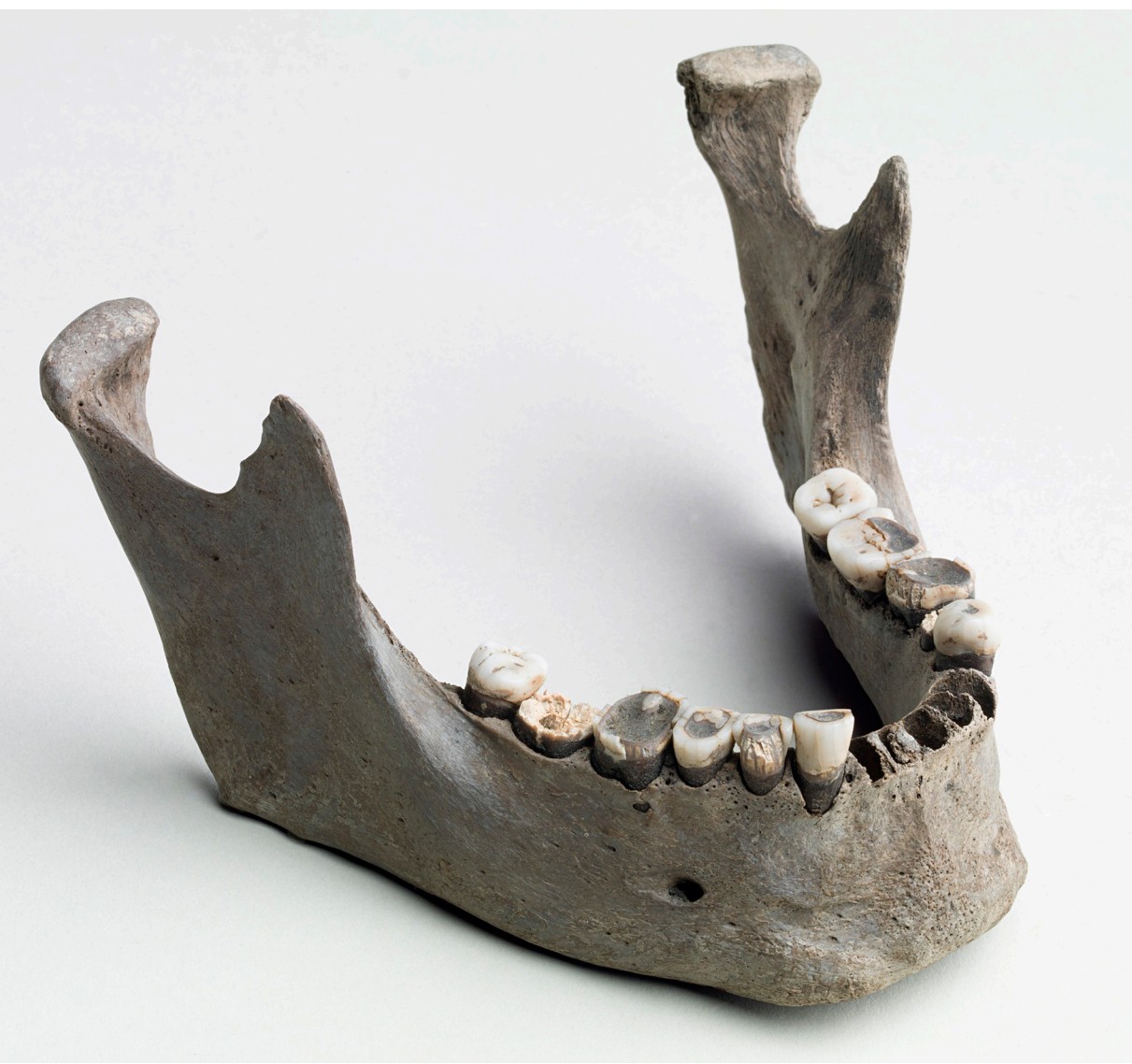

**Fig 4. The jawbone of Vittrup Man.** Several of the teeth are corroded due to soil chemical processes. All are clearly worn as a result of chewing. The character of the root impressions (alveolar sockets) from the front teeth indicate that these were lost after death–possibly when the jaw was brought to the light of day by a peat digger's spade. Photo: Arnold Mikkelsen.

## Analytical methods and results

An array of bioarchaeological methods has been applied to the skeletal remains of Vittrup Man for the purpose of gaining insight into his chronological position, genetic and geographic origin, and dietary life-history. As clarified below for each of the proxies, most of the presented data are exclusive to this study, whereas some derive from previously published data [1], that are being re-analysed and re-interpreted here to uncover new important details. Whether initially presented herein or elsewhere these analyses all presupposed destructive sampling. We have, however, been able to limit the interventions to a degree that the skeletal remains still appear complete to the eyes of visitors at Vendsyssel Historical Museum (Denmark), where the bones of this person are currently on display. The two teeth that were sacrificed to intensive sampling for DNA, AMS and isotopic analyses were documented in high-resolution photos. In

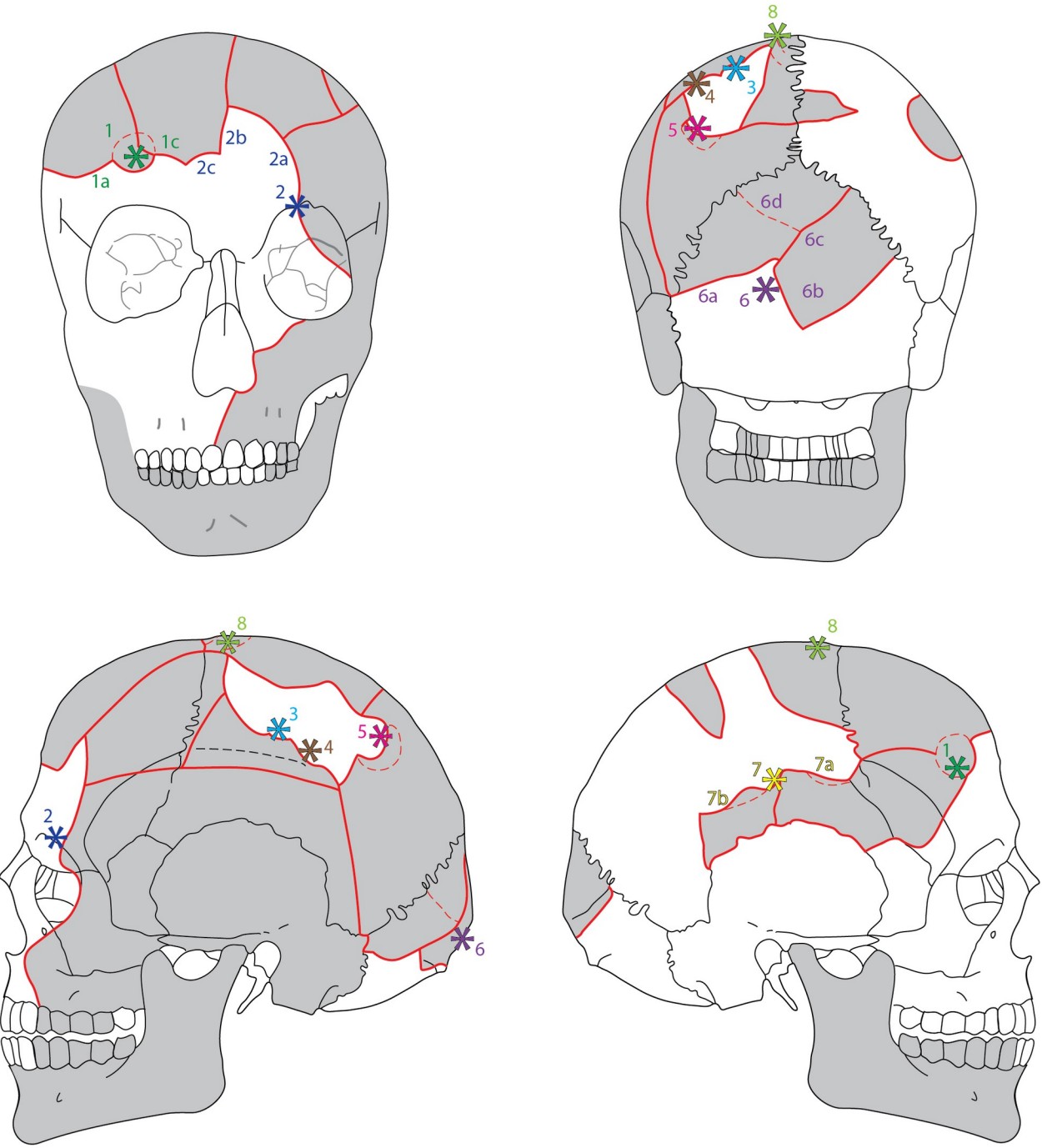

**Fig 5. Principal drawing of the cranium with indications of eight impact lesions and radiating fracture lines.** Parts that have been available for the present study are shown in grey. * indicate hit points. Lines in bold red represent fully completed fractures, whereas stippled red lines indicate incomplete fractures. Graphic drafted by Marie Louise Jørkov, Anders Fischer and Sidsel Wåhlin, graphically finalised by Rich Potter, University of Gothenburg.

addition, one of these teeth was run through a high-resolution 3D scan (SI.4 in S1 File) before cutting it up for incremental $\delta^{13}C$ and $\delta^{15}N$ analysis.

The bones of aurochs and domestic cattle mentioned below and presented in detail in Supplementary Information (SI.5 in S1 File) were likewise photo-documented before destructive

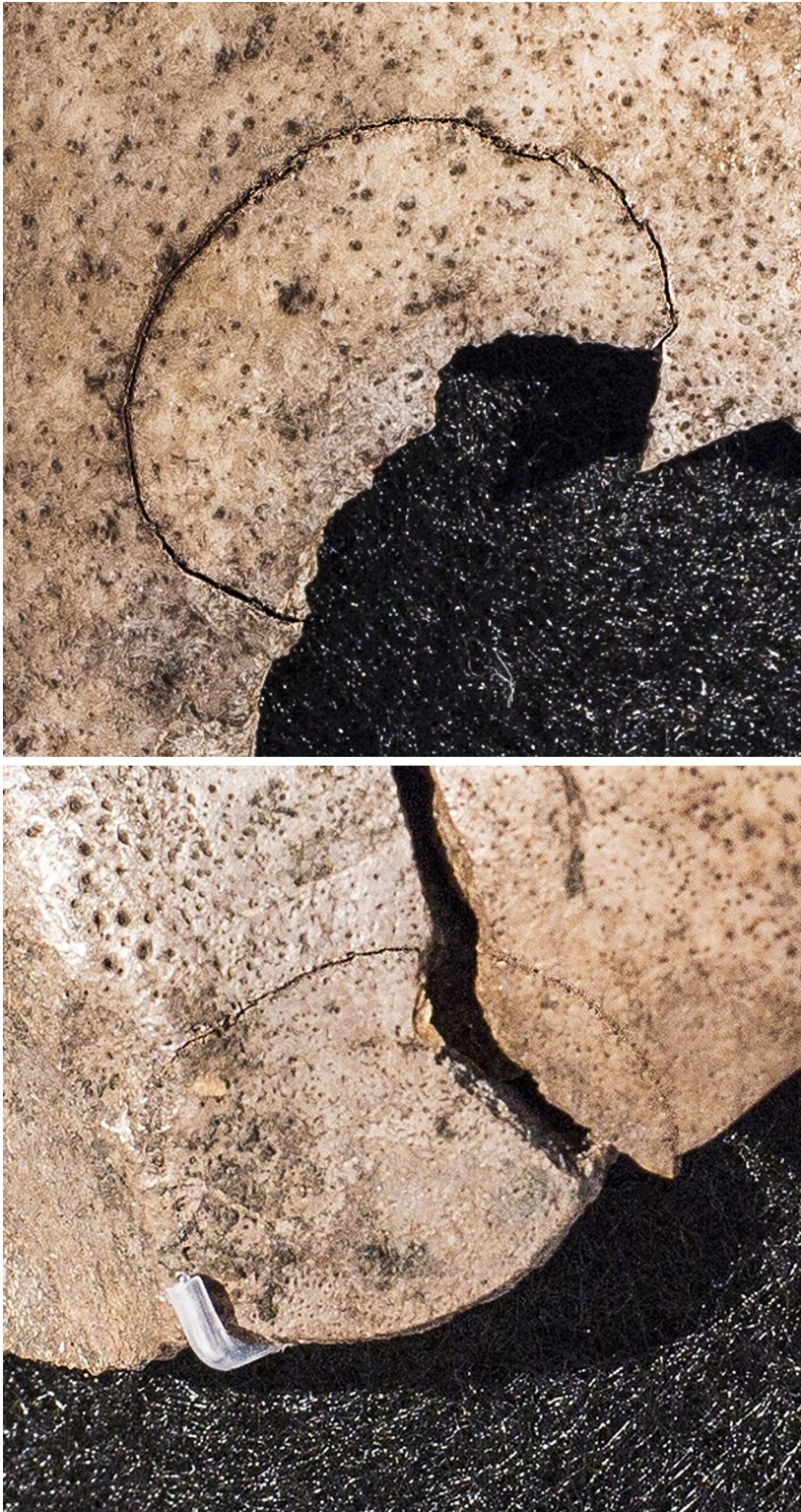

**Fig 6. Areas of impact with oval fractures, the maximum dimensions of which are c. 2 cm.** Photos: Stephen Freiheit.

sampling for AMS dating and isotopic analysis. Due to the scarcity of skeletal elements available from Vittrup Man we have imposed upon ourselves a modesty in sampling, which may very well have prevented us from further, interesting details of his unique life-history.

## AMS dates

An AMS date with associated dietary isotope values from the human skeletal remains of Vittrup Man has already been published [1]. As part of the present project, additional six [14]C dates have been produced, based on human and animal bones from the locality (Table 1). In between the dates of human remains no statistically significant differences are seen. We, therefore, assume they all derive from one and the same individual. Taking contemporaneity for granted, the best estimate on the age of the skeleton is 4513±19 BP (uncal).

Before turning this date into calibrated calendar years, we have to consider if a marine reservoir effect is at play. Estimating this we use a simple linear relation between the $\delta^{13}C$ endpoint values of -21 ‰ (terrestrial) and -10 ‰ (marine), based on many measurements on local Stone Age fauna, already published. Using the recently calculated value of 273±18 years for the marine reservoir offset at a Neolithic site in the Danish archipelago coeval to Vittrup Man, our rough estimate on marine reservoir effect for the Vittrup Man combined AMS date is 22 years. Calibrated with OxCal v4.4 and the Intcal20 calibration curve, the absolute age is 3340–3097 cal. BC, 95.4%. If we instead use the standard value for the north Atlantic, i.e. 400 years, we arrive at a reservoir correction of 33 years. This gives a calibrated date of 33455–3029 cal. BC (95.4%), with 89.4% of the probability within the 3345–3082 cal. BC interval. Since the Vittrup site faces the North Sea, this latter date is used here. Consequently, Vittrup Man belongs to the

**Table 1. AMS dates and associated dietary carbon and nitrogen isotope ratios for human and cattle bones from the Vittrup river mire.**

| Material dated | Age in [14]C years BP | Reservoir corrected cal age BC (95.4%) | $\delta^{13}C$ (‰) | $\delta^{15}N$ (‰) | Atomic C: N | Collagen % | Lab. No | VHM inventory No. |
|---|---|---|---|---|---|---|---|---|
| *Vittrup, human bones* | | | | | | | | |
| *Palatinum* sin. | 4464±52 | 3336–2921 | -20.2 | 9.9 | 3.2 | 11.7 | UBA-29904 | 11600C |
| *Palatinum* sin. | 4565±29 | 3363–3102 | -20.0 | 10.6 | 3.2 | 10.5 | UBA-39121 | 11600C |
| *Tibia* sin. | 4498±43 | 3347–2947 | -20.0 | 9.5 | 3.2 | 7.3 | UBA-39549 | 11600A |
| *Talus* dex. | 4476±32 | 3335–2935 | -20.2 | 9.8 | 3.2 | 14.4 | UBA-39550 | 11600B |
| Combined date | 4513±19 | 3353–3102 | -20.1 | 10.0 | | | All above | 11600A-C |
| Combined and reservoir corrected date | 4480±19 | 3345–3029 | | | | | | |
| *Vittrup, bovid bones* | | | | | | | | |
| *Metacarpus*, aurochs | 4365±34 | 3092–2902 | -22.6 | 4.8 | 3.2 | 12.6 | UBA-40444 | 11599A |
| *Phalanx* I, aurochs | 4457±40 | 3342–2936 | -22.4 | 4.9 | 3.2 | 12.6 | UBA-40445 | 11599B |
| *Aurochs combined* | 4404±26 | 3262–2917 | | | | | | |
| *Metacarpus*, domestic cattle (?) | 4735±30 | 3632–3377 | -20.9 | 5.6 | 3.2 | 9.6 | UBA-40446 | 11602A |

UBA-29904 and 39121 are from different parts of one and the same bone. The find circumstances for this skull fragment and the human tibia and talus imply they may very well derive from one and the same individual, which would accord with the radiocarbon results. Combination of the four human dates is statistically acceptable (X2 test: df = 3, T = 5.6 (5% 7.8)). The two dates for aurochs bones may possibly come from the same animal (X2 test; df = 1, T = 3.1 (5% 3.8)). An overlap in time with the human is possible, but not likely since 94% of the probability is within the 3101–2917 cal BC range (see Fig SI.1.6 in S1 File).

epoch of the Neolithic Funnel Beaker Culture (FBC), which in Denmark spans the interval of time c. 3900–2800 cal. BC.

## Genetic ancestry

Genomic ancient DNA data from Vittrup Man was published in [1], where it was shown to share close ancestry with roughly coeval Hunter-Gatherer (HG) individuals of Pitted Ware Culture (PWC) association from the Swedish island of Gotland. This contrasted to the findings from the majority of Danish Neolithic individuals from the FBC epoch, which display the typical Early European Farmer ancestry with minor proportions of admixture from Western European Hunter-Gatherer groups (conventional geneticist terminology, no a priori connotation as to actual mode of subsistence) [1]. To contribute further nuances and details on his ancestry, we re-analysed genomic data from Vittrup Man along with the genome of the Svinninge Vejle male (SI.12 in S1 File), in the context of 16 Gotlandic PWC associated individuals published in Coutinho et al. 2020 and not included in [1].

Principal Component Analysis (PCA) based on previously published genomic data from 288 ancient European Hunter-Gatherer and Farmer individuals shows (Fig 7, adapted from

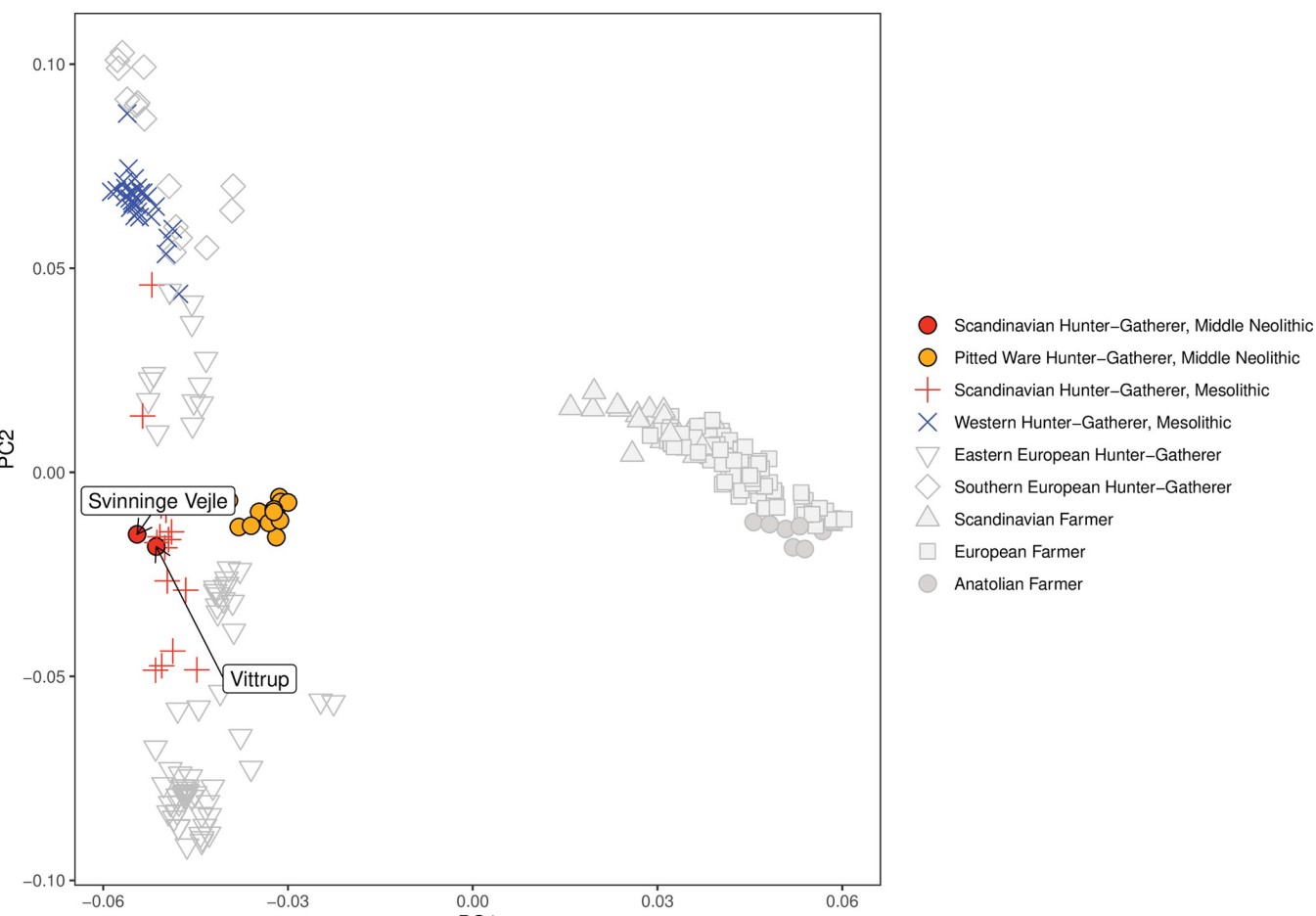

**Fig 7. Principal Component Analysis (PCA) projecting Vittrup Man, Svinninge Vejle and PWC individuals onto a reference panel of 288 ancient individuals from European Hunter-Gatherer and Neolithic Farmer groups.** The Vittrup and Svinninge Vejle individuals cluster with humans of Mesolithic age, found in Norway and Sweden.

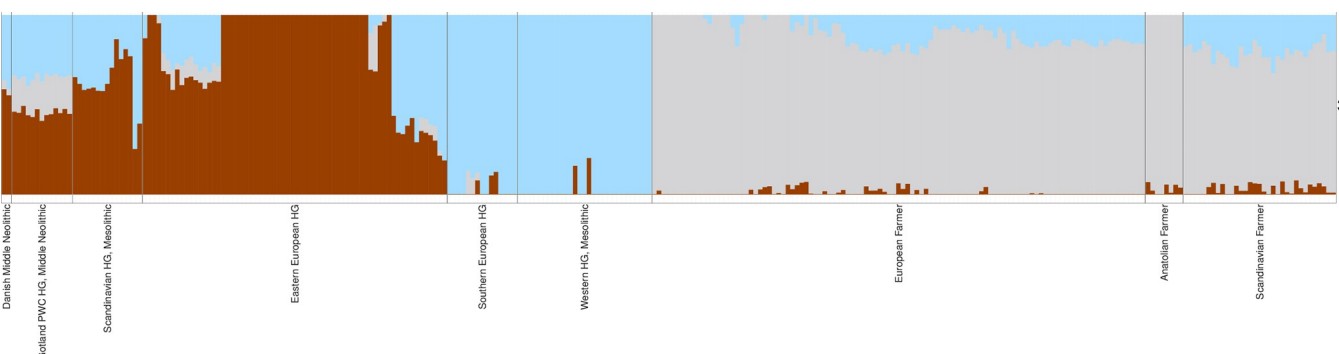

**Fig 8. Ancestry proportions for Vittrup Man and Svinninge Vejle (the two leftmost bars) in a context of 288 ancient individuals from European Hunter-Gatherers and Neolithic Farmers, inferred using ADMIXTURE with K = 3 ancestral components.**

Allentoft et al. [1]) that both Vittrup Man and Svinninge Vejle cluster closely with Mesolithic age Norwegian and Swedish Hunter-Gatherers. The Gotlandic PWC associated individuals also fall close to the Norwegian and Swedish HGs and Vittrup Man, but with a shift towards the Neolithic Farmers—consistent with previously reported presence of Early European Farmer ancestry in those individuals [2].

Results from a model-based clustering analysis (ADMIXTURE) mirror those from the PCA. The Vittrup and Svinninge Vejle individuals show most similar ancestry proportions to Mesolithic age Norwegian and Swedish Hunter-Gatherers, but distinct from the local Western European HGs present in Denmark during the preceding millennia (Fig 8). The Gotlandic PWC individuals carry a similar composition of Hunter-Gatherer ancestry, but with an increased contribution from a component that is maximised in Early European Farmer individuals. The minor 'grey' ancestry contributions to the Vittrup and Svinninge Vejle individuals, seen in Fig 8, could indicate potential gene flow from a local farmer population. We investigated this question using D-statistics in the form of D (Swedish HG, Vittrup/Svinninge Vejle; Early European Farmers, YRI), showing no evidence of significant levels of Farmer ancestry. Therefore, we can at least rule out the possibility of recent genetic admixture between the ancestors of Vittrup and Svinninge Vejle and Farmers with Anatolian ancestry. The larger grey contributions in the Gotlandic humans of PWC association represent genuine admixture with Scandinavian Neolithic Farmers (Fig SI.6.2 in S1 File). In summary, the genetic results suggest that the Vittrup and Svinninge Vejle individuals migrated to Denmark from a region not affected by Neolithic Farmer ancestry, presumably an area to the north of the known distribution of the Funnel Beaker Culture.

## Polygenic trait predictions

We aimed to use polygenic score predictions to obtain approximate inferences of physical appearance and physiology of Vittrup Man (SI.7 in S1 File). However, low genome coverage prevents us from deriving meaningful inferences for this individual. Instead, we obtained the polygenic scores for the chronologically and genetically closely-related individual from Svinninge Vejle, as computed in [1, 27]. Here, a high DNA preservation quality (genome coverage ~1X) allows–after imputation [1] –to produce polygenic scores for skin, hair and eye colour, height and basal metabolic rate. The scoring method is based on using a gene-trait association study of present-day inhabitants of the United Kingdom (UK Biobank, [28]). The considerable distance in time and ancestry between the present-day panel and the ancient individual implies that polygenic trait inferences can only be made with large reservations (SI.7 in S1 File).

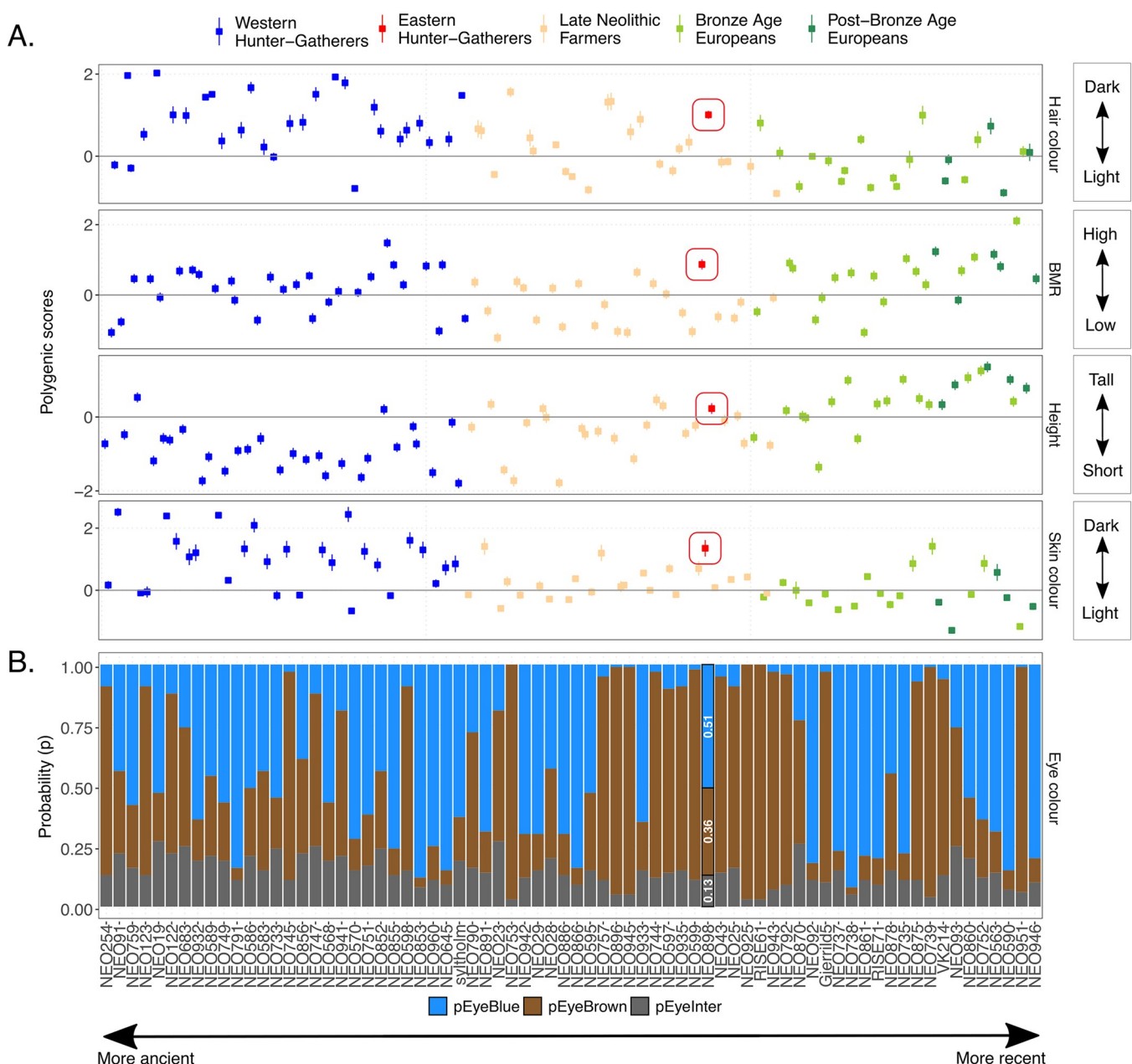

**Fig 9. Polygenic trait predictions for the Svinninge Vejle individual (figure computed using polygenic scores obtained from [1]).** A: Polygenic scores for the Danish samples ordered chronologically, for four traits of interest: hair colour, basal metabolic rate (BMR), height and skin colour. B: Probability of having blue vs. brown vs. intermediate eye colour (green/hazel/grey). The Svinninge Vejle individual is highlighted via inserted probability values. The error bars denote 95% credible intervals.

When comparing the polygenic scores of Svinninge Vejle with his contemporaneous individuals from Denmark [27], we find that—based on these scores alone—he may not have appeared remarkably different from them. As can be seen from Fig 9A, upper panel, he is predicted to have had somewhat darker skin, but his predicted genetic height and relatively dark hair colour appear to have been within the range of those predicted for his local European Neolithic Farmer contemporaries (Fig SI.7.2 in S1 File). He is also predicted to have probably had blue eyes (Fig 9B) and a relatively high basal metabolic rate—equivalent to the amount of

energy that the human body needs to perform its most basic functions. Thus, polygenic score predictions indicate that the Svinninge Vejle individual would have looked somewhat different from the majority of historic period or present-day Scandinavians, particularly in being relatively short and having darker skin and hair.

## Strontium and oxygen isotopes indicate a childhood far away from Vittrup

Strontium ($^{87}Sr/^{86}Sr$) and oxygen isotope ($\delta^{18}O$) analysis was performed on enamel samples from the upper left first premolar (UP1; +4) and the lower left third molar (M3, 'wisdom tooth'; -8). The crown of the former was formed during Vittrup Man's years 2–7, whereas the latter was formed during years c. 9 to 13–14 [29] (see SI.2 in S1 File). The strontium values are 0.7134 and 0.7159, respectively (Table SI.8.1 in S1 File). These values are out of range compared to the other genetically profiled humans from the Funnel Beaker Culture epoch in Denmark (Fig 10 and [1]). Values below 0.709 and above 0.712 are suspected to represent non-locals to the south Scandinavian landscapes [14, 30, 31], to which the find spot of Vittrup Man belongs. Strontium isotope analysis of three bones of local terrestrial mammals, chronologically closely connected to Vittrup Man, confirms his non-local provenience (Table SI.8.1, cf. SI.1 and SI.5 in S1 File).

Determining the place of origin with strontium isotopes is difficult because a number of places may have similar baseline values. Since the DNA results suggest a geographic origin for Vittrup Man on the Scandinavian Peninsula or neighbouring islands, we focus on that region. The ratios found in his dentition in fact occur in several places, including coastal parts of NW Norway and SW Sweden [13, 32–34]. The higher value from his third molar as compared with the premolar suggests a change of residence before the age of 9–12 years (SI.8 in S1 File).

The carbonate $\delta^{18}O$ value from Vittrup Man's premolar (-5.9‰ VPDB) is more negative than most observed in early prehistoric humans from Denmark, while the value from his third molar (-4.6‰ VPDB) is close to the mean of 150 Neolithic and Bronze age individuals from Denmark (-4.4‰ VPDB) (SI.8 in S1 File). This is consistent with a residential change from a colder, i.e. more northern part of Scandinavia, to a warmer region.

## Dietary isotopes reveal a chequered life-history

From several bones and teeth of Vittrup Man, stable isotope ratios of carbon ($\delta^{13}C$) and nitrogen ($\delta^{15}N$) were measured, aiming at elucidating his food composition during various periods of his lifetime (SI.8–SI.10 in S1 File). The measurements were conducted on two different materials: collagen (from bone and dentine), and enamel ($\delta^{13}C$ only). The former primarily reflects the composition of the protein part of the food, whereas the latter represents the whole diet [35–37]. The dentine of the lower left wisdom tooth was sampled incrementally, in order to follow dietary changes during the years of its formation.

Each sample represents a span of time. In the case of dentine, almost no remodelling takes place after its formation. Bones, on the other hand, are continuously remodelled at a pace depending on physical activity level, health, nutritional stress, biological age and sex [38, 39]. The longest intervals of time represented in our samples from Vittrup Man can be expected for the cortical bones (*Tibia* and *Talus)*, which may represent as much as two decades prior to death. For the tooth increments, we estimate the intervals to represent time slices in the order of one year or less. In all cases, samples may, however, include re-modelled body tissue reflecting the composition of diet earlier in life [40, 41].

We measured $\delta^{13}C$ in two enamel samples and obtained values of -10.5‰ from the premolar and -14.4‰ from the third molar (Table SI.8.2 in S1 File). The value for the former is unusual for Neolithic individuals from Denmark, whereas the value for the latter is common

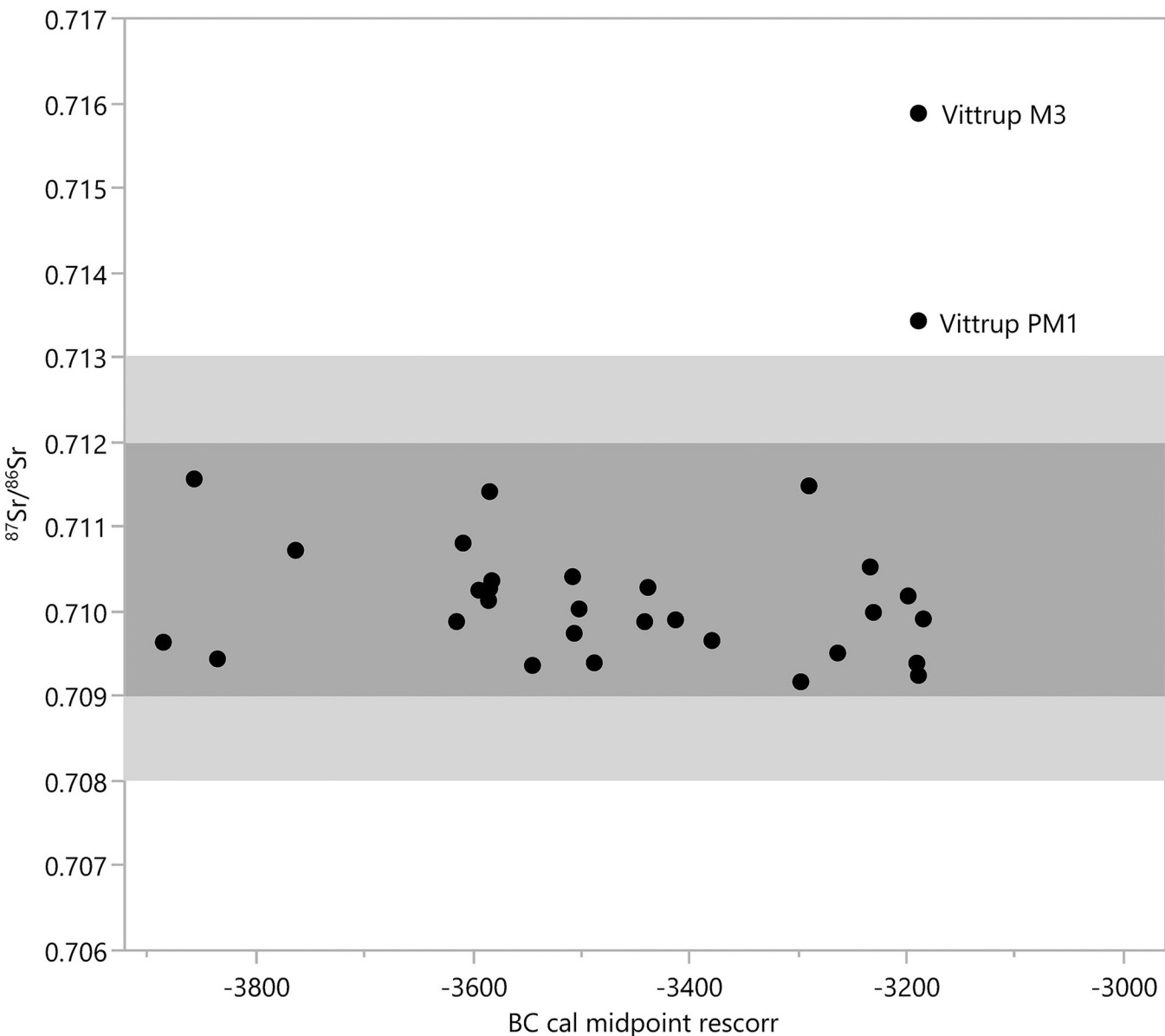

**Fig 10. Strontium isotope values of Danish human skeletal remains of the Funnel Beaker Culture epoch for which a genomic DNA profile is available.**
Vittrup Man plots clearly outside the Danish baseline range, indicated as grey zones. The highly dissimilar values for the two of his teeth analysed indicate a geographical change of residence. Data from [1] and SI.8 in S1 File this study.

for humans of this period and geographic zone [37] (see SI.8 in S1 File). Either way, these $\delta^{13}C$ values suggest a considerable consumption of marine foods in early childhood (2–7 years after birth) and a shift to a diet more dominated by resources from terrestrial and/or freshwater environments before the age of 13–14 years. In combination, they imply he lived a fisher-hunter-gatherer way of life in a coastal region during his earliest years.

Vittrup Man's bone collagen $\delta^{13}C$ and $\delta^{15}N$ values plot centrally amongst the measurements for Danish skeletons from the Funnel Beaker epoch (Fig 11; [42–44]). Consequently, in his adulthood when the collagen in these bones was formed, his food composition did not differ from what was normal among his contemporaries in the region where he died. His diet was

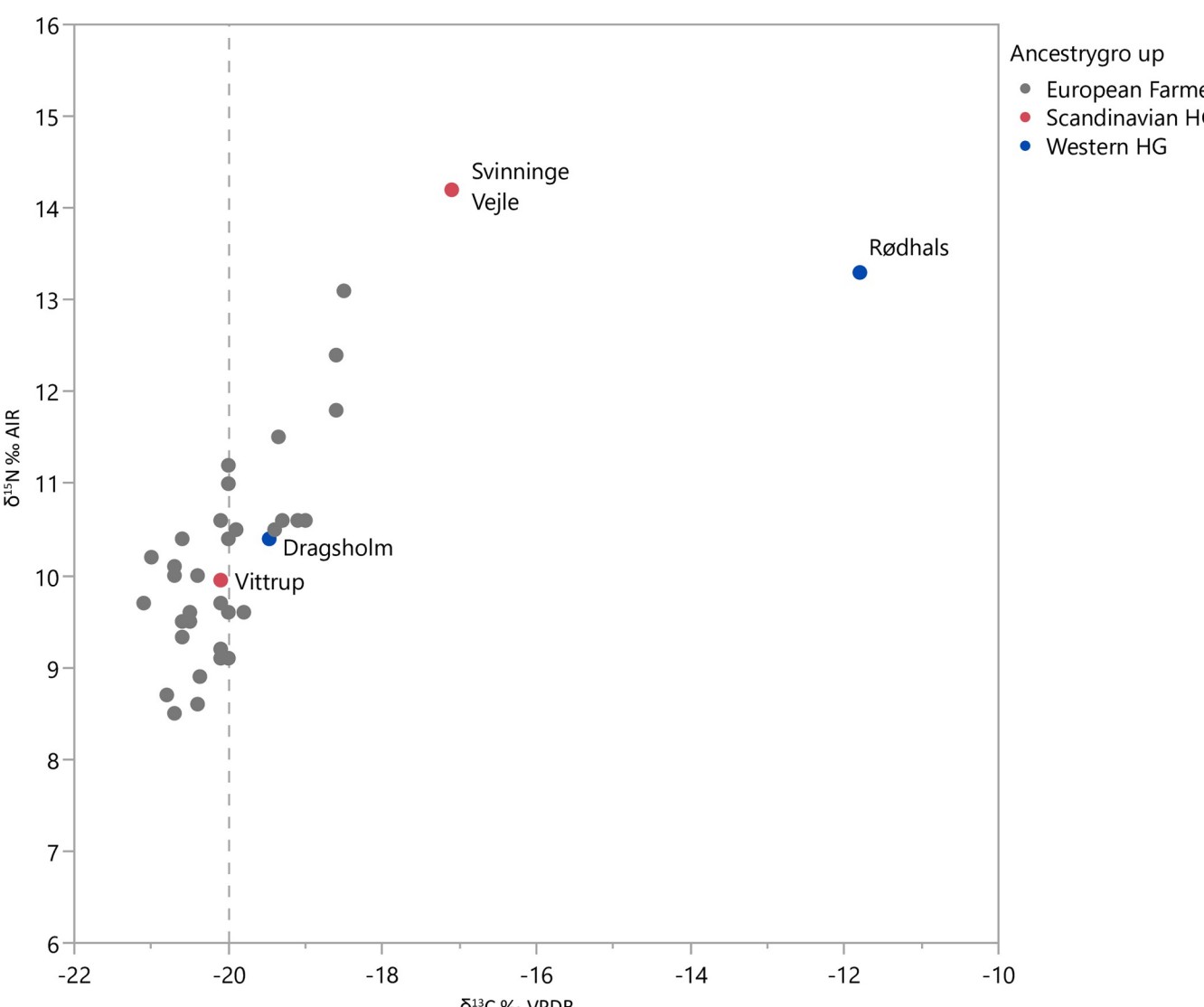

**Fig 11. Dietary and genetic characteristics of humans from the Funnel Beaker epoch in Denmark for which there is a genomic profile.** Individuals with a foreign genetic signature are marked in colour. All others (grey) have a predominantly European Farmer-derived genetic signature. For the individuals to the right of the dashed line at $\delta^{13}C$ = -20‰, sea food was probably a regular part of their diet. Individuals to the left of the line, such as Vittrup Man, consumed dominantly terrestrial sources. Data from [1].

most likely based on a combination of cereals, milk and/or herbivore meat, possibly with a minor supplement of high trophic level marine ingredients (SI.9 in S1 File).

A series of dentine samples from Vittrup Man's lower left wisdom tooth provides us with a chronologically insight into a part of his lifetime characterised by a significant change in his way of life (Fig 12, Fig SI.10.1 and Table SI.10.1 in S1 File). This is thanks to the roughly incremental nature of the dentine samples, formed approximately during the age interval of 12 to 20 years (SI.2 in S1 File), implying that the isotopic values represent averages of foods consumed within intervals of time in the order of half a year. The combination of $\delta^{13}C$ and $\delta^{15}N$ ratios from the tooth root suggest that the diet of his teens would have been dominated by a combination of freshwater fish and products from terrestrial sources (cf. SI.9 in S1 File). The synchronous movements of the $\delta^{13}C$ and $\delta^{15}N$ dentine profiles (Fig 12) indicate only slight

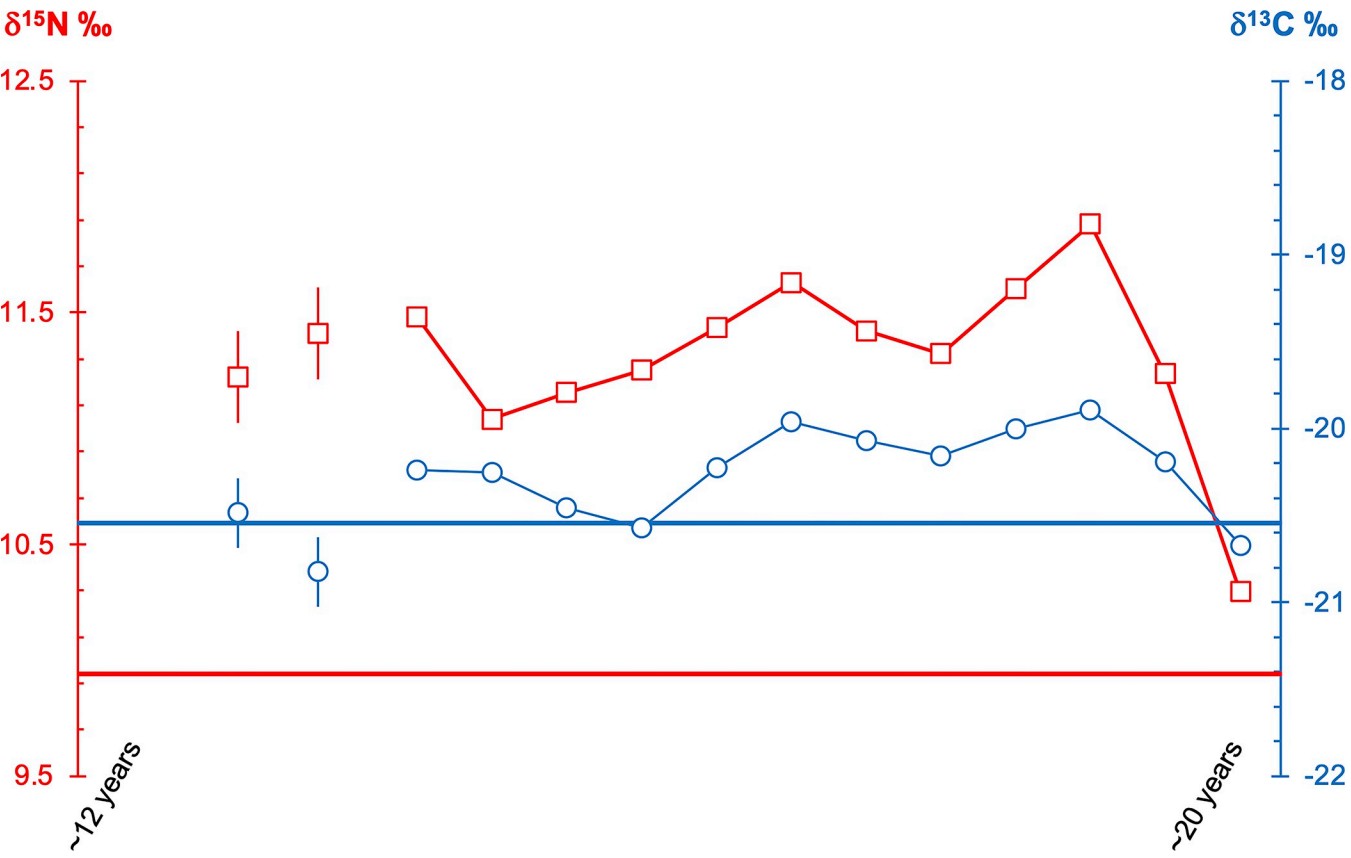

**Fig 12. Dietary change of Vittrup Man during major parts of his lifetime as indicated through a chronologically ordered series of dietary isotopic ratios.** The teenage years are represented in detail via incremental values (dots and triangles) from a wisdom tooth, formed during his years c. 12–20 (horizontal axis from left to right). In the lower part of the diagram the blue and red horizontal lines show the isotopic values for his palate, representing the years prior to death in his thirties. The significant fall in his late teenage years detected in the two incremental profiles may reflect a change from inland fisher-hunter-gatherer to farmer. The analytical error (+/- 0.2‰ 1sd) is shown on the leftmost two samples.

trophic level increases and decreases on half-a-year basis–low values representing larger proportions of plant food. This dietary regime came to an end at the close of his teenage years, when the $\delta^{13}$C and $\delta^{15}$N tooth incremental isotope values simultaneously dropped, approaching the values seen in his bone collagen. The radical fall in $\delta^{15}$N (half a trophic level) and the simultaneous fall in $\delta^{13}$C reflects a major change in food composition with vegetable resources probably getting a more pronounced role. It would suggest a change of his way of life from inland fisher-hunter-gatherer to farmer.

## Calculus analysis documents actual foods consumed

To most present-day humans of age, plaque is a familiar phenomenon, as was it in prehistory. Over a human's lifetime, it can result in the formation of dental calculus, a complex calcified bacterial biofilm rich in DNA and proteins [45], that can survive for millennia after the death of the individual. Palaeoproteomic analysis of dental calculus can be informative for subsistence studies since it can in some cases provide complementary information to dietary stable isotope analyses by identifying the specific animal and plant taxa consumed. This is exemplified by our palaeoproteomic data for Vittrup Man (SI.3 in S1 File), indicating the consumption of species from a variety of aquatic and terrestrial environments.

We identified ovine collagen but are unable to determine whether it derives from goat (*Capra hircus*) or sheep (*Ovis aries*) (Table SI.3.1 in S1 File). Moreover, we have indications for the presence of phocids (true seals) and cetaceans (whales) in Vittrup Man's diet. The former is represented by the harbour seal (*Phoca vitulina*). Which cetacean specie(s) is (are) represented in the latter group cannot be determined (Table SI.3.1 in S1 File). Hunting such huge fauna offshore requires specialised maritime equipment and techniques. Although remains of truly sea-worthy vessels suitable for offshore trips and open sea whale hunting appear still to be ill-represented in the archaeological record for Stone Age Scandinavia [46], evidence of tuna fishing and whaling coeval to Vittrup Man is well documented for southern Norway [47].

Collagen from cod (*Gadus morhua*) is identified in Vittrup Man's calculus (Table SI.3.1 in S1 File). Depending on the biological age and thus size, this species can be caught in shallow coastal waters or in the open sea [48]. It is frequently found at Late Mesolithic sites in Denmark [49–52], whereas fish remains are generally less well-represented in Funnel Beaker assemblages of this country [44].

A potential contribution of marine foods in Vittrup Man's calculus is interesting. Most likely this residue was mainly formed late in life (SI.2 in S1 File) when he subsisted on a diet typical of individuals associated with the Danish Funnel Beaker Culture. Based on measurements of carbon and nitrogen stable isotopes in human bones it has for long been discussed to what degree (if any) aquatic resources were part of the diet among Neolithic individuals from Denmark [44, 53–61]. The proteomic analysis indicates that members of this population engaged in open sea fishing and even had whales on their menu.

## Discussion

### Three contrasting cultural and genetic environments

Vittrup Man dates to c. 3300–3100 cal. BC (Table 1 and Fig SI.1.6 in S1 File), which in a Danish context translates into the era of the Funnel Beaker Culture, more precisely the first half of what is locally termed Middle Neolithic A [62]. Consequently, the relation between farmers and foragers revealed in this study has nothing to do with the local neolithization, c. 3900 cal. BC, with its initial influx of European Farmers and the associated disappearance of local Western European Hunter-Gathers, often dealt with in literature. On the contrary, we present a far less debated stage, when farmers were already well established and expressed themselves with a row of large-scale local social manifestations, representing economic surplus and cultural bloom. The construction of the numerous architecturally impressive passage graves peaked around that time [63, 64]. Likewise, this period is characterised by a high level of activity in wetland deposition of humans, domestic animals, pottery, flint axes, etc. [65–68]. Moreover, the products of FBC potters in that period reached remarkable artistic heights [69, 70].

According to current research, the Funnel Beaker Culture was probably the sole archaeological culture represented in present-day Denmark at the time of Vittrup Man (SI.12 in S1 File). More than half a millennium had passed since the disappearance of the genetically indigenous Mesolithic Western European Hunter-Gatherers (Fig 13). The last traces known from this population are from the skeletal remains of Rødhals Man and Dragsholm Man [1] and from DNA recovered from a piece of chewed tar from Syltholm [71]. However, Vittrup Man is heralding a new era of genetic Hunter-Gatherer presence in this region, as evidenced by material culture representing the Pitted Ware Culture (PWC). Rooted on the Scandinavian Peninsula, the PWC manifested itself along the coasts of northern and northeastern Denmark during parts of its existence [2, 13, 72–75]; cf. SI.12 in S1 File [2, 13, 72–75]. The earliest evidence of the PWC known from Denmark is from Djursland, where it co-occurred with FBC material culture from c. 3100 cal. BC [76]. Approximately 200–250 years would pass before the

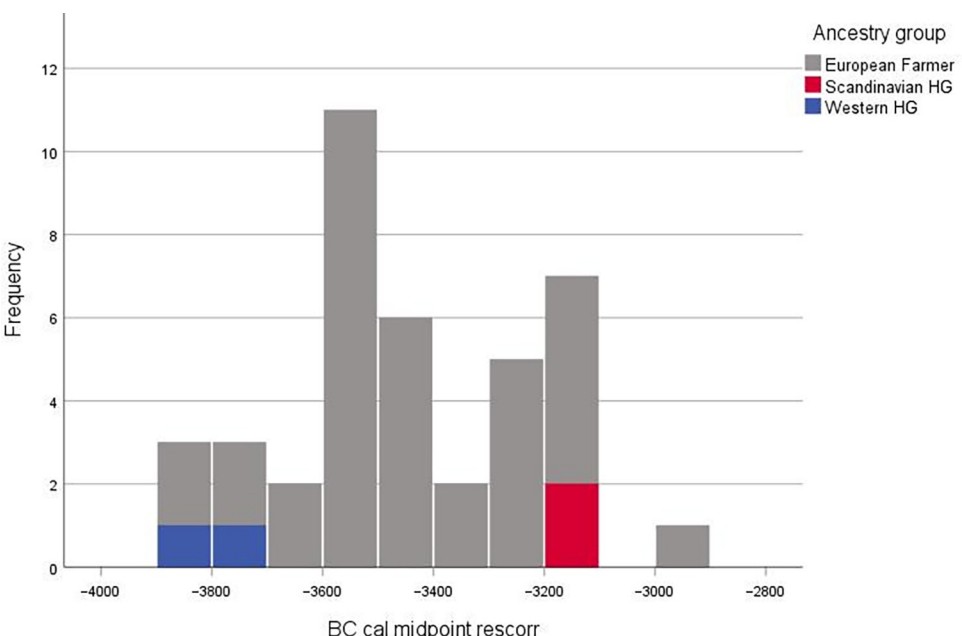

**Fig 13. Chronological distribution of genetically profiled human skeletal remains from the epoch of the Funnel Beaker Culture in Denmark.** Blue represents genetically Western Hunter Gatherers (late survivors of the local Mesolithic population: Rødhals Man and Dragsholm Man). Red: individuals of Scandinavian Hunter-Gatherer ancestry (Vittrup and Svinninge Vejle). Grey: individuals of European Farmer ancestry. Data from [1].

Single Grave Culture/Corded Ware Culture, originating in central and south-eastern Europe, made its arrival around 2850 cal. BC [1, 77, 78].

At the time of Vittrup Man, people carrying Anatolian ancestry inhabited all of Denmark as well as (at least) the southern and southwestern parts of Sweden (Fig 2, grey dots). In the previous centuries—during the local Early Neolithic—this population was also permanently present further north, on the island of Gotland and in lowland areas of the Scandinavian Peninsula well beyond Stockholm and the Oslo Fjord. This is evidenced by finds of material culture, such as megalithic burial monuments, settlements and wetland depositions with funnel beaker pottery [79–81].

North of the areas inhabited by Funnel Beaker groups during the period of their maximal northward geographical expansion there will have been humans that were closely related to the Norwegian and Swedish individuals of Mesolithic age known to us (red + symbols in Fig 2) [1]). However, genomic data is lacking from Norway and Sweden for members of this population. The reason for this is probably the unfavourable preservation conditions for Stone Age human bones that pertain to the generally lime-deficient soils found on most of the Scandinavian Peninsula.

Around the transition from the Early Neolithic to early Middle Neolithic, the FBC population abandoned northern parts of its previously inhabited territories on the Scandinavian Peninsula and nearby islands which were then occupied by PWC groups [72]. These groups practised farming to a minor extent [73], and were highly dependent on marine resources, including seal and whale [47, 57, 82, 83]. Nuclear genomic data for PWC individuals, found in the lime-rich soils of the island of Gotland, show admixture between Scandinavian Hunter-Gatherers and European Farmers [2].

Thus, at the time of Vittrup Man southern Scandinavia was inhabited by at least three populations with different economies and genetic ancestries: to the south, FBC groups with

European Farmer ancestry; to the north, hunter-gatherers with a purely local Mesolithic ances-try; and in between, PWC groups characterised by mixed Scandinavian Hunter-Gatherer and European Farmer ancestry. The genetic make-up of the latter population reveals that close physical contacts between hunter-gatherer and farmer groups were involved. Based on the lim-ited DNA evidence available, offspring of such contacts were common among fisher-hunters, but not among farmers.

## A foreigner from across the sea

The combined results of our carbon, nitrogen, oxygen and strontium stable isotope analyses of Vittrup Man reveal a chequered life-history. He began as a fisher-hunter-gatherer in a northern Scandinavian coastal region, which could, as an extreme possibility, be along the Norwegian coast near the Arctic Circle (SI.8 and SI.13 in S1 File). Then came a change in diet to one domi-nated by resources from terrestrial and/or freshwater environments. Before the age of 18–19 years, he ended up among regular farmers, possibly within the Funnel Beaker cultural environ-ment in the northwestern end of Denmark, where his life came to an end several years later.

Genetically, Vittrup Man's closest parallel from Neolithic Denmark is a human from Svin-ninge Vejle (Figs 2 and 7; SI.12 in S1 File). These two males deviate significantly from coeval individuals associated with the Pitted Ware Culture on Gotland [2, 84, 85]. Instead, they share genetic ancestry with Mesolithic individuals from the Scandinavian Peninsula. Unfortunately, the genetic composition of PWC individuals outside Gotland is unknown, but it is likely that admixture with European Farmers occurred also in these areas, where material culture reflect-ing FBC presence was more widespread than observed in the archaeological record of Gotland.

Based on our combined set of genetic and isotopic data, we infer that the individuals from Vittrup and Svinninge Vejle most likely had biological relatives in societies on the Scandina-vian Peninsula, outside the core regions of the FBC—i.e., north of the upper border of Fig 2. From these regions, there is a rich archaeological record of coeval coastal and inland settle-ment [86–92], but currently no genomic data for the period in question. Alternatively, it may be argued that these two individuals originated from confined groups of hunter-gatherers liv-ing further south while being surrounded by FBC-associated societies. However, it is difficult to point out traces of any such remnant groups of genetically indigenous SHGs [13, 93–96].

According to polygenic scores, the Svinninge Vejle individual had a relatively high basal metabolic rate. Such a physiological characteristic accords with the northern geographic ori-gin, deduced on the basis of our genomic, strontium and $\delta^{18}$O analyses. This characteristic would most likely have made him (and possibly also Vittrup Man) cope slightly better with cold conditions than the contemporaneous European Farmers, who made up the majority of the population in Denmark and southern Sweden (Fig 2) [97].

Regardless of Vittrup Man's original 'port of departure', his arrival in NW Denmark indi-cates extensive travel by boat. The most direct route from the west coast of Sweden would have required crossing >75 km of open sea, much of the time out-of-sight of land, even if using the islands of Læsø or Anholt as stopovers. Crossing the shortest distance from Norway would imply a sea-crossing of well over 100 km. Under most weather conditions, a safe voyage across these waters would require a genuinely seaworthy vessel and clever seamanship. According to inferences based on traditional archaeological sources, the capability of conducting such sea journeys was probably well established at that time among the Funnel Beaker Culture as well as among Pitted Ware groups and coastal foragers even further north [5, 13, 46, 81, 98–101] and SI.13 in S1 File. This interpretation may now find support in our proteomic results, which indicate that he consumed food, procured through open sea fishing, sealing and whaling (SI.3 in S1 File). With his background as a child and juvenile among fisher-hunter-gatherers Vittrup

Man may have had competencies in maritime activities acknowledged within the TRB group he lived with as an adult.

## A violent death in a wider perspective

Why should this foreigner—somewhere between c. 3300 and 3100 cal. BC—be deposited in a bog far distant from his place of birth? The fatal blows to Vittrup Man's head exclude accidental drowning. He could be a victim of feud or murder. The thousands of coeval depositions of animals, humans and precious items in similar environmental settings in Denmark and southern parts of Sweden make it more probable that he was sacrificed.

For generations, archaeologists have been aware of traces of Pitted Ware activity in Denmark, primarily along the coasts [5, 7, 12, 13, 102]. These finds gave inspiration to a hypothesis of seasonal visits by boat from Pitted Ware groups in Norway and Sweden in search of flint [5]. Several publications have subsequently stressed the involvement of the PWC in trade relations that resulted in the movement of items, not least flint axes, from the flint-rich areas in Denmark and Scania to Pitted Ware contexts on the Scandinavian peninsula [6, 13, 75, 103 Taf. 21, 1, 104]. Possibly less known is the existence of a return-flow to northern Jutland of exotic items produced in coastal Norway (SI.13 in S1 File). This inter-Scandinavian exchange system functioned centuries before and continued centuries after the life of Vittrup Man. During his life span the flint axes traded were of the thin-butted type—a very common artefact in Denmark. It is also recorded by the thousands in Sweden with a dense distribution that reaches north of the large lakes partly seen in Fig 2 [31, 72, 105 Abb. 314, 106 Fig 22.22]. Additionally, the axe type is found in considerable numbers in the Oslo Fjord region while more scattered, beyond the apparent limits of FBC habitation, along the Norwegian coast further west and north [107–109] (SI.13 in S1 File).

When combining previous archaeological knowledge with the genetic, chronological and isotopic information for the Vittrup and Svinninge Vejle individuals now available, a reciprocal relation between Denmark and the Scandinavian Peninsula north of Scania seems to transpire: valuable artefacts moved north and humans moved south. Other resources may also have changed hands [110, 111], and several alternative explanations as to the driving forces of such an exchange system may be suggested. In the specific case of Vittrup Man, two main scenarios could explain his life story. He might have been a person of Scandinavian Peninsula origin, involved in flint exchange between northern Jutland and Norway/Sweden, who settled down in NW Denmark and became integrated into the local FBC as an equal member of his new society. Another possibility is that he was taken prisoner, possibly far north in west coast Scandinavia, and spent his years of maximal physical strength within a NW Danish FBC community as a captive and source of labour (SI.14 in S1 File).

Anyhow, we stress that the roles of northern foragers and Pitted Ware Culture associated groups in the exchange system may well have been active ones, facilitated by a mobile, coastal lifestyle. Within this perspective, the PWC groups may be seen as middlemen, responsible for a coastal exchange system transporting goods, and possibly humans, between Middle or North Scandinavia and Denmark.

That Vittrup Man was sacrificed does not necessarily have any bearing on his social status. Historic sources as well as ethnography mention slaves in relatively high social roles [112, 113], and present cases where high-status, highly valued persons were chosen for sacrifice [114].

## Conclusions

Vittrup Man stands out as a genetically distinct foreigner in Neolithic Denmark. His ancestry is similar to Mesolithic individuals from the Scandinavian Peninsula and shows no significant

levels of admixture with Neolithic farmer-related ancestry. Additionally, he is the only published Funnel Beaker period individual from Denmark with a clearly non-local strontium signature. Moreover, $\delta^{13}C$ and $\delta^{18}O$ values in his teeth hold evidence of a childhood in a coastal environment far to the north. In combination, these observations indicate that his place of birth was on the Scandinavian Peninsula, possibly at the Norwegian coast next to the Arctic Circle. Additionally, his dietary isotope values indicate that during his teenage years, he switched from a fisher-hunter-gatherer diet to that of a farmer, and during the last many years of life his diet was similar to that of the majority of contemporary individuals found in Denmark. Thus, during his years of maximal physical labour capacity, Vittrup Man lived in a farming society, probably hundreds and possibly more than a thousand kilometres from his childhood home. Many explanations for such a drastic change in life-style and geography are possible, as exemplified in ethnographic and early historic sources. He may have been an immigrant or trader who became integrated into equal social standing as other members of the local Funnel Beaker society. He could also have been a captive/slave providing labour and possibly maritime skills. However, neither his way of death nor his lifestyle allows definite conclusions as to his social standing.

The circumstances of Vittrup Man's death appear easily explained. At that time, it was common practice in present-day Denmark to sacrifice humans in bogs, and these acts were often conducted in violent ways. Evidently, such dubious honour was also given to persons of non-local provenience.

## Supporting information

**S1 File. SI.1 –SI.14.**
(PDF)

**S2 File. Deposaries/reproducibility.**
(DOCX)

## Acknowledgments

We acknowledge a conservation specialist from Finland (NLES, previous employee of Bevaringscenter Nordjylland) for calling attention to the archaic appearance of the Vittrup skull; Paula Reimer (14CHRONO, Queen's University Belfast), for providing methodological clarifications on AMS dates and isotopic measurements; Paul Fullagar (University of North Carolina-Chapel Hill) and David Dettman (University of Arizona, Tucson) for effective handling of our enamel Sr and $\delta^{13}C$ & $\delta^{18}O$ samples, respectively, during a complicated period of pandemic close downs; Poul Otto Nielsen (Danish National Museum) for access to archives and for assistance in having artefact photos produced for the paper. Moreover, we express our sincere thanks to Niels Lynnerup (Copenhagen University), Poul Otto Nielsen (Danish National Museum) and Kurt Gron (Durham University) for useful comments on earlier versions of the manuscript.

## Author Contributions

**Conceptualization:** Anders Fischer.

**Data curation:** Anders Fischer, Karl-Göran Sjögren, Theis Zetner Trolle Jensen, Marie Louise Jørkov, Gabriele Scorrano, T. Douglas Price, Darren R. Gröcke, Morten E. Allentoft.

**Formal analysis:** Karl-Göran Sjögren, Marie Louise Jørkov, Tharsika Vimala, Alba Refoyo-Martínez, Gabriele Scorrano, Andrés Ingason.

**Funding acquisition:** Darren R. Gröcke, Eske Willerslev, Morten E. Allentoft, Kristian Kristiansen.

**Investigation:** Anders Fischer, Marie Louise Jørkov, Gabriele Scorrano, Darren R. Gröcke, Anne Birgitte Gotfredsen, Lasse Sørensen, Verner Alexandersen, Jesper Stenderup, Ole Bennike, Andrés Ingason.

**Methodology:** Anders Fischer, Karl-Göran Sjögren, Theis Zetner Trolle Jensen, Gabriele Scorrano, Martin Sikora, Morten E. Allentoft.

**Project administration:** Anders Fischer, Morten E. Allentoft, Kristian Kristiansen.

**Resources:** Anders Fischer, Per Lysdahl, Sidsel Wåhlin, Jesper Stenderup, Eske Willerslev.

**Supervision:** Anders Fischer, Martin Sikora, Fernando Racimo, Morten E. Allentoft.

**Validation:** Anders Fischer, Theis Zetner Trolle Jensen.

**Visualization:** Anders Fischer, Karl-Göran Sjögren, Theis Zetner Trolle Jensen, Marie Louise Jørkov, Tharsika Vimala, Alba Refoyo-Martínez, Gabriele Scorrano, Lasse Sørensen, Sidsel Wåhlin, Ole Bennike.

**Writing – original draft:** Anders Fischer, Karl-Göran Sjögren, Theis Zetner Trolle Jensen, Marie Louise Jørkov, Per Lysdahl, Tharsika Vimala, Alba Refoyo-Martínez, Gabriele Scorrano, T. Douglas Price, Anne Birgitte Gotfredsen, Lasse Sørensen, Verner Alexandersen, Kristian Kristiansen.

**Writing – review & editing:** Anders Fischer, Karl-Göran Sjögren, Theis Zetner Trolle Jensen, Marie Louise Jørkov, Per Lysdahl, Tharsika Vimala, Alba Refoyo-Martínez, T. Douglas Price, Darren R. Gröcke, Anne Birgitte Gotfredsen, Ole Bennike, Andrés Ingason, Rune Iversen, Martin Sikora, Fernando Racimo, Morten E. Allentoft, Kristian Kristiansen.

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
