## [Decision Letter · Decision Letter 0]

28 Jul 2023

PONE-D-23-20645Vittrup Man – the life-history of a genetic foreigner in Neolithic DenmarkPLOS ONE

Dear Dr. Fischer,

Thank you for submitting your manuscript to PLOS ONE. After careful consideration, we feel that it has merit but does not fully meet PLOS ONE’s publication criteria as it currently stands. Therefore, we invite you to submit a revised version of the manuscript that addresses the points raised during the review process.

We look forward to receiving your revised manuscript.

Kind regards,

Peter F. Biehl, PhD

Academic Editor

PLOS ONE

Journal Requirements:

2. We noted in your submission details that a portion of your manuscript may have been presented or published elsewhere. "This submission is a spin-off of a manuscript submitted to ‘Nature’ in September 2022, and now in the process of minor adjustment after positive peer review. It is available on bioRxive: Population Genomics of Stone Age Eurasia (biorxiv.org). 

- Parts of the data analyzed in this PlosOne submission's SI.6 and presented in its main text Figs 7 and 8 were taken from above mentioned Nature manuscript. Our re-analysis with the inclusion of data from another published paper (not made use of in mentioned Nature manuscript) resulted in a fundamentally new understanding of North European Neolithic population history. Further clarification is presented in the main text's Introduction and in SI.6.

- All data analyzed in this submission's SI.7 and presented in its main text Fig 8 were taken from above mentioned Nature  manuscript and from Irving-Pease EK, Refoyo-Martínez A, Ingason A. The Selection Landscape and Genetic Legacy of Ancient Eurasians. bioRxiv. 2022. The graphic presentation Main text Fig. 8 and the graphics in SI.7 are produced specifically for the current submission, and serve as the basis for the written texts associated with them. For further clarification please check this submission's SI.7." Please clarify whether this conference proceeding or publication was peer-reviewed and formally published. If this work was previously peer-reviewed and published, in the cover letter please provide the reason that this work does not constitute dual publication and should be included in the current manuscript.

3. We note that Figures 2 in your submission contain map images which may be copyrighted. All PLOS content is published under the Creative Commons Attribution License (CC BY 4.0), which means that the manuscript, images, and Supporting Information files will be freely available online, and any third party is permitted to access, download, copy, distribute, and use these materials in any way, even commercially, with proper attribution. For these reasons, we cannot publish previously copyrighted maps or satellite images created using proprietary data, such as Google software (Google Maps, Street View, and Earth). For more information, see our copyright guidelines: http://journals.plos.org/plosone/s/licenses-and-copyright.

a. You may seek permission from the original copyright holder of Figures 2 to publish the content specifically under the CC BY 4.0 license.  

4. We note that Figures 1,4 and 6 in your submission contain copyrighted images. All PLOS content is published under the Creative Commons Attribution License (CC BY 4.0), which means that the manuscript, images, and Supporting Information files will be freely available online, and any third party is permitted to access, download, copy, distribute, and use these materials in any way, even commercially, with proper attribution. For more information, see our copyright guidelines: http://journals.plos.org/plosone/s/licenses-and-copyright.

a. You may seek permission from the original copyright holder of Figures 1,4 and 6 to publish the content specifically under the CC BY 4.0 license. 

5. We note that Figures in your Supporting Information contain copyrighted images. All PLOS content is published under the Creative Commons Attribution License (CC BY 4.0), which means that the manuscript, images, and Supporting Information files will be freely available online, and any third party is permitted to access, download, copy, distribute, and use these materials in any way, even commercially, with proper attribution. For more information, see our copyright guidelines: http://journals.plos.org/plosone/s/licenses-and-copyright.

a. You may seek permission from the original copyright holder of Figures to publish the content specifically under the CC BY 4.0 license. 

Additional Editor Comments:

Please address all comments in detail before resubmitting.

Reviewers' comments:

Reviewer's Responses to Questions

**Comments to the Author**

1. Is the manuscript technically sound, and do the data support the conclusions?

Reviewer #1: Yes

Reviewer #2: Yes

2. Has the statistical analysis been performed appropriately and rigorously? 

Reviewer #1: Yes

Reviewer #2: Yes

3. Have the authors made all data underlying the findings in their manuscript fully available?

Reviewer #1: Yes

Reviewer #2: Yes

4. Is the manuscript presented in an intelligible fashion and written in standard English?

Reviewer #1: Yes

Reviewer #2: Yes

5. Review Comments to the Author

Reviewer #1: Overview

This is an excellent paper well suited to PLoS ONE. As promised in the title, the authors successfully present a combined biomolecular, archaeological and anthropological study. While we have long expected that contacts existed between HGs and farmers at the level of specific individuals, identifying these has proven very difficult and has only become possible with advances in aDNA over the last decade. Vittrup Man presents an fascinating case study demonstrating the level of detail that is now possible.

Most of the comments below are relatively minor, but some issues are raised. There could be more synthesis between some of the specialist studies (e.g., the dental palaeopathology and the proteomics). There is a question mark over some of the sequential dentine results, which may be impacted by humic contamination, and the untreated premolar d18O and d13C values should probably be adjusted based on the treated third molar offset. One suggestion that I would strongly recommend to the authors is that – if access is possible – the M1 be sequentially sampled to provide more information on early life diet and included in the paper. The claim for a coastal HG diet early in life is currently based only on a d13C enamel value on a premolar, but collagen results would be more convincing and provide more detail on the inferred marine diet.

Comments

26/ “The deadly maltreated body of Vittrup Man…”

Grammatically this doesn’t make sense.

38/ “Such a volatile life-history”

Certainly a dramatic life change, but not sure that ‘volatile’ is the most appropriate word here.

131/ suggest decomposing in place of ‘rotting’

132/ “Incomplete salvage of bones actually present in the peat dig should also be considered”

Suggest ‘incomplete recovery’

The ‘weighted mean’ in Table 1 should include the results of the standard chi-square test to demonstrate the compatibility of the 14C dates.

171/ states that the 14C dates for the two aurochs bones are statistically different and so must come from different animals. Despite the appearance of the plot in Figure SI.1.6 ,using R_combine in OxCal returns: X2-Test: df=1 T=3.1(5% 3.8), which is not rejected, i.e., they are not inconsistent with the same age. See also discussion of this in SI. The combined date is later than the dates for Vittrup Man.

177/ Just as likely that there is no detectable marine protein in the diet of this individual, though the date in minimally affected in any case.

195/ Some additional context should be provided for Svinninge Vejle man

218/ “Therefore, we can at least rule out the possibility of recent genetic admixture between the ancestors of Vittrup and Svinninge Vejle and Farmers.”

Suggest phrasing this to be very clear that reference is to groups with Anatolian ancestry rather than the shorthand ‘Farmers’ which could be taken out of context.

245/ “and a relatively high basal metabolic rate - equivalent to the amount of energy that the human body needs to perform its most basic functions.”

More explanation needed here as to why this is of interest, i.e., that this differs from what has been found in those with Anatolian farmer ancestry (as stated later in the paper, line 441-4).

261/ Would be useful to note in the text the sample size for FBC individuals with strontium isotope values

264/ “Strontium isotope analysis of three bones of local terrestrial mammals”

Sr isotopes in bones are very likely to be contaminated with Sr from the the depositional environment.

278/ Since the d18O sample for the PM was not pre-treated, and a difference of -0.67‰ was found in a comparison of untreated and pre-treated M3 samples, the implication is that the PM value should be even lower, ca. -6.6‰. Should note in main text that these values are relative to VPDB.

286/ “The measurements were conducted on two different materials: collagen (from bone and dentine), and enamel.”

As the previous sentence refers to both d13C and d15N, should be clear here that only d13C was measured on the enamel, as it could be read to mean that both were measured.

297/ As for d18O, there seems to be a significant offset between the untreated and pre-treated M3 for d13C. The implication is that the d13C value would be ca. 0.9‰ more negative.

297-303/ The enamel d13C results are used to identify a “fisher-hunter gatherer way of life in a coastal region”. But it is the collagen results that should show this more clearly as they provide predominantly a protein signal. Why was this tooth not analysed sequentially to confirm and add further information (d15N) to the enamel result? Was the entire root used for DNA? That seems excessive. The M1 root would be the most appropriate tooth to investigate diet during early life; this would provide a valuable addition to the paper and it is strongly recommend that this is done.

362/ “The proteomic analysis indicates that members of this population engaged in open sea fishing and even had whales on their menu.”

If so, it was not sufficiently frequent to impact either their adult bone collagen or their adolescent dentine stable isotope values.

422/ An origin near the Arctic Circle seems unlikely given the distances involved. SI line 599 states “Due to coarse sampling, it cannot be excluded that areas with equivalently high ratios can also be found further south along the Norwegian coast.”

439/ What about the eastern Baltic as another potential source for Vittrup Man to at least be mentioned? Many individuals from Zvejnieki, Latvia show WHG/SHG ancestry (as understood some years ago). This may be less compatible with the d18O results suggesting a colder climate, but this would also apply to some of the other scenarios considered here. Also, as the SI notes, a few colder years during formation of the enamel could account for the difference.

Jones, E.R., Zarina, G., Moiseyev, V., Lightfoot, E., Nigst, P.R., Manica, A., Pinhasi, R. and Bradley, D.G. 2016. The Neolithic transition in the Baltic was not driven by admixture with early European farmers. Current Biology 27: 1-7.

479/ “valuable artefacts moved north and humans moved south”

Presumably other things besides humans, as alluded to earlier. See also papers by Marek Zvelebil on this topic, e.g.,

Zvelebil, M. 1998. Agricultural frontiers, Neolithic origins, and the transition to farming in the Baltic Basin. In: M. Zvelebil, R. Dennell and L. Domanska (eds.), Harvesting the Sea, Farming the Forest: The Emergence of Neolithic Societies in the Baltic Region: pp. 9-27. Sheffield: Sheffield Academic Press.

505/ “Thus, during his years of maximal physical strength”

Relevance of this?

SI

Surprisingly, no link is made between the dental palaeopathological results and the calculus proteomic evidence for Treponema maltophilum

333/ “Thanks to AMS dating and the measuring of their strontium isotopic and dietary stable isotope (δ13C, δ15N and δ34S) values”

No δ34S results are presented in the paper.

Figure SI.6.2 labels Vittrup and Svinninge Vejle only as ‘Danish Middle Neolithic’ which is potentially confusing, as the whole point is that these are totally anomalous in the context of other Danish MN individuals.

648/ the pretreated d18O and d13C values are used for the M3 enamel, but no pretreatment was applied to the PM. As an offset was found, this implies that it should be applied to the PM which alters its values slightly.

697/ “probably due to amendment” Presumably ‘manuring’ is intended.

707/ The sequential results for the M3 suggest only limited dietary diversity during the period of its formation. There is no reference to this in the speculative claim made for the bone collagen results.

Table SI.10.1 – there seems to be a moderate but significant correlation between the d13C and C:N values that suggests the presence of humic contaminants affecting the lowest d13C values. Would consider omitting C:N values above 3.4, cf.

Fernández-Crespo, T., Snoeck, C., Ordoño, J., de Winter, N.J., Czermak, A., Mattielli, N., Lee-Thorp, J.A. and Schulting, R.J. 2020. Multi-isotope evidence for the emergence of cultural alterity in Late Neolithic Europe. Science Advances 6(4): eaay2169.

745/ “δ34S values for three roughly contemporary Bos sp. specimens recovered near to the Vittrup individual (main text Table 1; Fig SI.1.6), may in this regard provide important base-line data.”

Table 1 does not present any δ34S values, and they are not mentioned at all in the main text.

812/ can the authors be more explicit on how the marine reservoir correction was carried out for the date for Svinninge Vejle. What DR was used? This is presumably not 273±18 years, which refers to the offset between terrestrial and marine dates in the cited paper.

Reviewer #2: This is a very significant study that illustrates the dynamic nature of human interaction during the fourth millennium BC using multiple lines of evidence. It also demonstrates the suite of techniques that can be applied to the analysis of human skeletal materials, even those recovered over a century earlier. The most significant thing for me is that Vittrup Man dates to 3300-3100 BC, several centuries after the introduction of agriculture to southern Scandinavia. While this dating is certainly discussed in the text, I think it might be possible to highlight it a bit more sharply to make it stand out. One of the things I'm finding in recent studies (especially archaeogenetic) from southern Scandinavia is a compression of dated sites and materials from the different centuries of the fourth millennium BC into a synthetic narrative which blurs the distinction between the first centuries of the millennium and the second half. Vittrup Man's story is not part of the initial influx of European Farmer and disappearing WHG populations but rather a later development during the "Funnel Beaker climax," which is really fascinating in its own right.

6. PLOS authors have the option to publish the peer review history of their article (what does this mean?). If published, this will include your full peer review and any attached files.

Reviewer #1: No

Reviewer #2: No

---

## [Author Response · Author response to Decision Letter 0]

14 Nov 2023

26 August 2023

Author’s responses (also provided as an individual file with responces in red color).

PONE-D-23-20645

Vittrup Man – the life-history of a genetic foreigner in Neolithic Denmark

Dear Dr. Fischer,

Thank you for submitting your manuscript to PLOS ONE. After careful consideration, we feel that it has merit but does not fully meet PLOS ONE’s publication criteria as it currently stands. Therefore, we invite you to submit a revised version of the manuscript that addresses the points raised during the review process.

• A marked-up copy of your manuscript that highlights changes made to the original version. You should upload this as a separate file labeled 'Revised Manuscript with Track Changes'. Done; split into two files: main text and SI, respectively. 

• An unmarked version of your revised paper without tracked changes. You should upload this as a separate file labeled 'Manuscript'. Done. In two files: main text and SI, respectively.

We look forward to receiving your revised manuscript.

Kind regards, Peter F. Biehl, PhD, Academic Editor

PLOS ONE

Journal Requirements:

Done. 

2. We noted in your submission details that a portion of your manuscript may have been presented or published elsewhere. "This submission is a spin-off of a manuscript submitted to ‘Nature’ in September 2022, and now in the process of minor adjustment after positive peer review. It is available on bioRxive: Population Genomics of Stone Age Eurasia (biorxiv.org). 

- Parts of the data analyzed in this PlosOne submission's SI.6 and presented in its main text Figs 7 and 8 were taken from above mentioned Nature manuscript. Our re-analysis with the inclusion of data from another published paper (not made use of in mentioned Nature manuscript) resulted in a fundamentally new understanding of North European Neolithic population history. Further clarification is presented in the main text's Introduction and in SI.6.

- All data analyzed in this submission's SI.7 and presented in its main text Fig 8 were taken from above mentioned Nature manuscript and from Irving-Pease EK, Refoyo-Martínez A, Ingason A. The Selection Landscape and Genetic Legacy of Ancient Eurasians. bioRxiv. 2022. The graphic presentation Main text Fig. 8 and the graphics in SI.7 are produced specifically for the current submission, and serve as the basis for the written texts associated with them. For further clarification please check this submission's SI.7." Please clarify whether this conference proceeding or publication was peer-reviewed and formally published. If this work was previously peer-reviewed and published, in the cover letter please provide the reason that this work does not constitute dual publication and should be included in the current manuscript.

This publication has not been published or peer-reviewed elsewhere. The primary data bases for the manuscript are obtained by methods from isotope analysis, physical anthropology and archaeology, produced within the present study and reserved for publication in this manuscript. Additionally, it makes use – and gain new insights from – the re-analysis of genomic data from the following three publications:

1) Coutinho et al. The Neolithic Pitted Ware culture foragers …. Am J Phys Anthropol. 2020, 

2) Allentoft et al (submitted to Nature). Population Genomics of Stone Age Eurasia. bioRxiv. 2022. pp. 1–72. doi:10.1101/2022.05.04.490594, 

3) Irving-Pease et al. The Selection Landscape and Genetic Legacy of Ancient Eurasians. bioRxiv 2022.09.22.509027; doi: https://doi.org/10.1101/2022.09.22.509027. 

In our manuscript, genomic data from these three papers has been combined into a new genomic panel, which we use to derive specific insights about ‘Vittrup Man’, as well as with the previously unpublished archaeological and physical anthropological data, for which the manuscript accounts fully.

References are made in all cases where we include data from these manuscripts.

3. We note that Figures 2 in your submission contain map images which may be copyrighted. All PLOS content is published under the Creative Commons Attribution License (CC BY 4.0), which means that the manuscript, images, and Supporting Information files will be freely available online, and any third party is permitted to access, download, copy, distribute, and use these materials in any way, even commercially, with proper attribution. For these reasons, we cannot publish previously copyrighted maps or satellite images created using proprietary data, such as Google software (Google Maps, Street View, and Earth). For more information, see our copyright guidelines: http://journals.plos.org/plosone/s/licenses-and-copyright.

We require you to either (a) present written permission from the copyright holder to publish thisese figures specifically under the CC BY 4.0 license, or (b) remove the figures from your submission:

a. You may seek permission from the original copyright holder of Figures 2 to publish the content specifically under the CC BY 4.0 license. 

The map (Fig 2) was made using natural earth data, the terms of which are in the following link: https://www.naturalearthdata.com/about/terms-of-use/

No permission is needed to use Natural Earth. Crediting the authors is unnecessary. In the figure caption we have added reference to www.naturalearthdata.com

4. We note that Figures 1,4 and 6 in your submission contain copyrighted images. All PLOS content is published under the Creative Commons Attribution License (CC BY 4.0), which means that the manuscript, images, and Supporting Information files will be freely available online, and any third party is permitted to access, download, copy, distribute, and use these materials in any way, even commercially, with proper attribution. For more information, see our copyright guidelines: http://journals.plos.org/plosone/s/licenses-and-copyright.

a. You may seek permission from the original copyright holder of Figures 1,4 and 6 to publish the content specifically under the CC BY 4.0 license. 

Please upload the completed Content Permission Form or other proof of granted permissions as an "Other" file with your submission. Done.

5. We note that Figures in your Supporting Information contain copyrighted images. All PLOS content is published under the Creative Commons Attribution License (CC BY 4.0), which means that the manuscript, images, and Supporting Information files will be freely available online, and any third party is permitted to access, download, copy, distribute, and use these materials in any way, even commercially, with proper attribution. For more information, see our copyright guidelines: http://journals.plos.org/plosone/s/licenses-and-copyright.

a. You may seek permission from the original copyright holder of Figures to publish the content specifically under the CC BY 4.0 license. 

Please upload the completed Content Permission Form or other proof of granted permissions as an "Other" file with your submission. Done – as concerns photos and Figures SI.1.2, SI.1.3 and SI.12.2. 

The maps Fig SI.13.2 and SI.13.5 were made using data (political boundary, water course, inland water) acquired in 2009 from MapCruzin.com. Data er freely available. The home page states, however, that the originators will appreciate mentioning "MapCruzin.com" – although this is not mandatory, cf. https://mapcruzin.com/download-free-arcgis-shapefiles.htm

In the respective figure captions, we have added reference to MapCruzin.com. 

6. Please review your reference list to ensure that it is complete and correct. If you have cited papers that have been retracted, please include the rationale for doing so in the manuscript text, or remove these references and replace them with relevant current references. Any changes to the reference list should be mentioned in the rebuttal letter that accompanies your revised manuscript. Done.

If you need to cite a retracted article, indicate the article’s retracted status in the References list and also include a citation and full reference for the retraction notice. No retracted article cited.

Additional Editor Comments:

Please address all comments in detail before resubmitting.

Reviewers' comments:

Reviewer's Responses to Questions

Comments to the Author

1. Is the manuscript technically sound, and do the data support the conclusions?

Reviewer #1: Yes

Reviewer #2: Yes

2. Has the statistical analysis been performed appropriately and rigorously?

Reviewer #1: Yes

Reviewer #2: Yes

3. Have the authors made all data underlying the findings in their manuscript fully available?

Reviewer #1: Yes

Reviewer #2: Yes

4. Is the manuscript presented in an intelligible fashion and written in standard English?

Reviewer #1: Yes

Reviewer #2: Yes

5. Review Comments to the Author

Reviewer #1: Overview

This is an excellent paper well suited to PLoS ONE. As promised in the title, the authors successfully present a combined biomolecular, archaeological and anthropological study. While we have long expected that contacts existed between HGs and farmers at the level of specific individuals, identifying these has proven very difficult and has only become possible with advances in aDNA over the last decade. Vittrup Man presents an fascinating case study demonstrating the level of detail that is now possible. We thank a lot for the minute commenting, which we have found extraordinarily useful and which have certainly improved the manuscript.

Most of the comments below are relatively minor, but some issues are raised. There could be more synthesis between some of the specialist studies (e.g., the dental palaeopathology and the proteomics). We accept, and have expanded on this, for instance in SI.2 – see below.

There is a question mark over some of the sequential dentine results, which may be impacted by humic contamination Thank you. We have now omitted the sample in question from the graphs in our revised version of main text Figure 12.

, and the untreated premolar d18O and d13C values should probably be adjusted based on the treated third molar offset. Thanks once more. An additional sentence has been added – see below.

One suggestion that I would strongly recommend to the authors is that – if access is possible – the M1 be sequentially sampled to provide more information on early life diet and included in the paper. The claim for a coastal HG diet early in life is currently based only on a d13C enamel value on a premolar, but collagen results would be more convincing and provide more detail on the inferred marine diet. This is a very constructive suggestion, and it has caused additional remarks to our SI chapters 4 and 8, thank you. It is a bulls-eye hit on a topic we have been concerned about for a while, and where our conclusion has been (and still is): we don’t do it. There are three reasons for this conclusion, which we hereby list in the order that in our minds represent progressively importance: 

1) economic and labor costs. 

2) we do not possess sample remains of relevance, and it will take nearly a year to produce a new dietary incremental series – in case the owner of the skeleton will judge an additional sample application positively. 

3) research ethical reasoning (cf. SI.4); our study demonstrates the skeletal remains of Vittrup Man to represent the only first-generation migrant known from the Danish Stone Age; there are relatively few teeth available from Vittrup Man and we feel an obligation that as many as possible of these shall be at disposal for future generations of researchers, who will very likely possess methods, which can reveal information that is currently totally out of reach; additionally, the clarification, which can potentially be gained via an additional incremental series, would be a pleasant supplement to our study, but is not a strict necessity for the over-all narrative of our manuscript (Vittrup Man is a genetic foreigner originating somewhere on the Scandinavian Peninsula).

Comments

26/ “The deadly maltreated body of Vittrup Man…”

Grammatically this doesn’t make sense. Thank you. ‘deadly’ now replaced with ‘lethally’.

38/ “Such a volatile life-history”

Certainly a dramatic life change, but not sure that ‘volatile’ is the most appropriate word here. OK. ‘volatile’ now replaced with ‘variable’. 

131/ suggest decomposing in place of ‘rotting’ Thank you. Corrected accordingly.

132/ “Incomplete salvage of bones actually present in the peat dig should also be considered”

Suggest ‘incomplete recovery’ Thank you. Corrected accordingly.

The ‘weighted mean’ in Table 1 should include the results of the standard chi-square test to demonstrate the compatibility of the 14C dates. Accepted; X2 result added.

171/ states that the 14C dates for the two aurochs bones are statistically different and so must come from different animals. Despite the appearance of the plot in Figure SI.1.6 ,using R_combine in OxCal returns: X2-Test: df=1 T=3.1(5% 3.8), which is not rejected, i.e., they are not inconsistent with the same age. See also discussion of this in SI. The combined date is later than the dates for Vittrup Man. Thank you. Corrected, and a new Fig. SI.1.6 produced.

177/ Just as likely that there is no detectable marine protein in the diet of this individual, though the date in minimally affected in any case. Yes, any marine reservoir effect will affect the date very little, and will have no effect as to the over-all narrative of the paper. 

195/ Some additional context should be provided for Svinninge Vejle man. Thanks. We have briefly added some context information as well as a reference to SI.12, where the complete context data available for the Svinninge Vejle individual is presented. Reviewer’s comment has caused us to extend SI.12. As a result, further two authors have been added to the SI chapter, and consequently an additional authors have been added at the front page of the main text.

218/ “Therefore, we can at least rule out the possibility of recent genetic admixture between the ancestors of Vittrup and Svinninge Vejle and Farmers.”

Suggest phrasing this to be very clear that reference is to groups with Anatolian ancestry rather than the shorthand ‘Farmers’ which could be taken out of context. Thanks. Brief adjustment made.

245/ “and a relatively high basal metabolic rate - equivalent to the amount of energy that the human body needs to perform its most basic functions.”

More explanation needed here as to why this is of interest, i.e., that this differs from what has been found in those with Anatolian farmer ancestry (as stated later in the paper, line 441-4). We prefer sticking to the short expression her, and then pick up on the topic in the discussion (previously lines 441-4), which we have now extended.

261/ Would be useful to note in the text the sample size for FBC individuals with strontium isotope values We do not possess exact data on this, but the size of our Sr samples have systematically been excessively large. We have added a reference to the Allentoft et al (submitted) paper, where any specifics as to the samples from FBC individuals are published. 

264/ “Strontium isotope analysis of three bones of local terrestrial mammals”

Sr isotopes in bones are very likely to be contaminated with Sr from the the depositional environment. OK. In the additional caption to Table SI 8.1 we have now added: “The risk of contamination from the depositional environment, potentially applying to the bone samples, is considered unproblematic, since it will also be local.”

278/ Since the d18O sample for the PM was not pre-treated, and a difference of -0.67‰ was found in a comparison of untreated and pre-treated M3 samples, the implication is that the PM value should be even lower, ca. -6.6‰. Should note in main text that these values are relative to VPDB. The treatment of the PM is unknown, so this is not relevant. VPDB added in the main text.

286/ “The measurements were conducted on two different materials: collagen (from bone and dentine), and enamel.”

As the previous sentence refers to both d13C and d15N, should be clear here that only d13C was measured on the enamel, as it could be read to mean that both were measured. OK. Fixed.

297/ As for d18O, there seems to be a significant offset between the untreated and pre-treated M3 for d13C. The implication is that the d13C value would be ca. 0.9‰ more negative. Treatment for the PM is unknown, see above.

297-303/ The enamel d13C results are used to identify a “fisher-hunter gatherer way of life in a coastal region”. But it is the collagen results that should show this more clearly as they provide predominantly a protein signal. Why was this tooth not analysed sequentially to confirm and add further information (d15N) to the enamel result? Was the entire root used for DNA? That seems excessive. The M1 root would be the most appropriate tooth to investigate diet during early life; this would provide a valuable addition to the paper and it is strongly recommend that this is done. Regrettably, yes: the total tooth root was used for the DNA analysis. This happened at a very early stage of the project, and admittedly was probably excessive.

362/ “The proteomic analysis indicates that members of this population engaged in open sea fishing and even had whales on their menu.”

If so, it was not sufficiently frequent to impact either their adult bone collagen or their adolescent dentine stable isotope values. It is a matter of ongoing debate to what degree (if any) marine food was part of diet among the majority of the population associated with the Funnel Beaker Culture in Denmark – as can be seen from our references 27,57–65. A final say on this may not be reached on the basis of current standard measurements of d13C and d15N in collagen. This is why we at the close of SI. 9 briefly mention other categories of archaeological evidence of importance to this discussion. 

422/ An origin near the Arctic Circle seems unlikely given the distances involved. SI line 599 states “Due to coarse sampling, it cannot be excluded that areas with equivalently high ratios can also be found further south along the Norwegian coast.” To us such a distant place is in fact within the likely possibilities. Within the present study we, however, prefer not to go into further discussion – and ask for the accept of the peer reviewer and the editor that we stick to the expression with the addition that it is “an extreme possibility”. 

439/ What about the eastern Baltic as another potential source for Vittrup Man to at least be mentioned? Many individuals from Zvejnieki, Latvia show WHG/SHG ancestry (as understood some years ago). This may be less compatible with the d18O results suggesting a colder climate, but this would also apply to some of the other scenarios considered here. Also, as the SI notes, a few colder years during formation of the enamel could account for the difference.

Jones, E.R., Zarina, G., Moiseyev, V., Lightfoot, E., Nigst, P.R., Manica, A., Pinhasi, R. and Bradley, D.G. 2016. The Neolithic transition in the Baltic was not driven by admixture with early European farmers. Current Biology 27: 1-7. We do not know of strictly comparable genomes outside the Scandinavian Peninsula, and the individuals from Zvejnieki were WHG, different from both Danish and Scandinavian HGs (cf. Allentoft et al (submitted to Nature). Population Genomics of Stone Age Eurasia. bioRxiv. 2022. pp. 1–72. doi:10.1101/2022.05.04.490594). Considering the very few genomes coeval to Vittrup Man that are currently available from northern Europe, we prefer not going into further discussion of such possibilities.

479/ “valuable artefacts moved north and humans moved south”

Presumably other things besides humans, as alluded to earlier. See also papers by Marek Zvelebil on this topic, e.g.,

Zvelebil, M. 1998. Agricultural frontiers, Neolithic origins, and the transition to farming in the Baltic Basin. In: M. Zvelebil, R. Dennell and L. Domanska (eds.), Harvesting the Sea, Farming the Forest: The Emergence of Neolithic Societies in the Baltic Region: pp. 9-27. Sheffield: Sheffield Academic Press. Accepted, and a brief addition to the text made.

505/ “Thus, during his years of maximal physical strength”

Relevance of this? Directly related to his social role in his new society. A small adjustment and a minor addition to the previous chapter made.

SI

Surprisingly, no link is made between the dental palaeopathological results and the calculus proteomic evidence for Treponema maltophilum. Good suggestion, thank you so much. We have expanded on this issue in SI.2 and have made several additional cross references, including one from SI.3 to SI.2.

333/ “Thanks to AMS dating and the measuring of their strontium isotopic and dietary stable isotope (δ13C, δ15N and δ34S) values”

No δ34S results are presented in the paper. References to the tables presenting these data have now been inserted.

Figure SI.6.2 labels Vittrup and Svinninge Vejle only as ‘Danish Middle Neolithic’ which is potentially confusing, as the whole point is that these are totally anomalous in the context of other Danish MN individuals. Aiming at clarification we have made minor adjustments of terminology in this SI chapter, including the caption to Fig SI.6.2.

648/ the pretreated d18O and d13C values are used for the M3 enamel, but no pretreatment was applied to the PM. As an offset was found, this implies that it should be applied to the PM which alters its values slightly. Agree. In fact, it is unknown whether a pretreatment was applied to the PM or not. We have added the following sentences to make sure this is stated more clearly: “From Table SI.8.2 it appears that an additional offset factor should possibly be suspected in between δ13C and δ18O values of unpretreated versus pretreated samples. However, offsets of the scale seen in the table will not affect our inferences below.” 

697/ “probably due to amendment” Presumably ‘manuring’ is intended. Yes, thank you. Adjusted accordingly.

707/ The sequential results for the M3 suggest only limited dietary diversity during the period of its formation. There is no reference to this in the speculative claim made for the bone collagen results. Reference now made.

Table SI.10.1 – there seems to be a moderate but significant correlation between the d13C and C:N values that suggests the presence of humic contaminants affecting the lowest d13C values. Would consider omitting C:N values above 3.4, cf.

Fernández-Crespo, T., Snoeck, C., Ordoño, J., de Winter, N.J., Czermak, A., Mattielli, N., Lee-Thorp, J.A. and Schulting, R.J. 2020. Multi-isotope evidence for the emergence of cultural alterity in Late Neolithic Europe. Science Advances 6(4): eaay2169. Sorry, we do not agree. All the samples have been treated the same. If there were humic acids in the first three samples that would mean humic acids should also be in the remainder of the dentine samples. There is a good correlation with other segments of the dentine and that is often found in teeth dentine. It does not mean humic acid. It is reflecting changes in the diet. The Fernández-Crespo et al. paper is only applicable to their samples - not widely across all samples and studies.

745/ “δ34S values for three roughly contemporary Bos sp. specimens recovered near to the Vittrup individual (main text Table 1; Fig SI.1.6), may in this regard provide important base-line data.”

Table 1 does not present any δ34S values, and they are not mentioned at all in the main text. OK. For the purpose of precision, in SI.11 we have now adjusted the sentence in question into “δ34S values (Table SI.11.1) for three roughly contemporary Bos sp. specimens recovered near to the Vittrup individual (main text Table 1; Fig SI.1.6),”. The second pair of brackets refers to the dates for these bones.

812/ can the authors be more explicit on how the marine reservoir correction was carried out for the date for Svinninge Vejle. What DR was used? This is presumably not 273±18 years, which refers to the offset between terrestrial and marine dates in the cited paper. We did not use a DR, only an offset value. We have added a brief sentence on this to the table caption.

Reviewer #2: 

This is a very significant study that illustrates the dynamic nature of human interaction during the fourth millennium BC using multiple lines of evidence. It also demonstrates the suite of techniques that can be applied to the analysis of human skeletal materials, even those recovered over a century earlier. 

The most significant thing for me is that Vittrup Man dates to 3300-3100 BC, several centuries after the introduction of agriculture to southern Scandinavia. While this dating is certainly discussed in the text, I think it might be possible to highlight it a bit more sharply to make it stand out. One of the things I'm finding in recent studies (especially archaeogenetic) from southern Scandinavia is a compression of dated sites and materials from the different centuries of the fourth millennium BC into a synthetic narrative which blurs the distinction between the first centuries of the millennium and the second half. Vittrup Man's story is not part of the initial influx of European Farmer and disappearing WHG populations but rather a later development during the "Funnel Beaker climax," which is really fascinating in its own right. Very good suggestion, thank you. As a consequence, we have expanded the initial part of the paper’s Discussion as follows: “the relation between farmers and foragers revealed in this study has nothing to do with the local neolithization, c. 3900 cal. BC, with its initial influx of European Farmers and the associated disappearance of local Western European Hunter-Gathers, often dealt with in literature. On the contrary, we present a far less debated stage, when farmers were already well established and expressed themselves with a row of large-scale local social manifestations, representing economic surplus and cultural bloom.”

6. PLOS authors have the option to publish the peer review history of their article (what does this mean?). If published, this will include your full peer review and any attached files.

Do you want your identity to be public for this peer review? For information about this choice, including consent withdrawal, please see our Privacy Policy.

Reviewer #1: No

Reviewer #2: No

---

## [Editor Report · Decision Letter 1]

28 Dec 2023

Vittrup Man – the life-history of a genetic foreigner in Neolithic Denmark

PONE-D-23-20645R1

Dear Dr. Fischer,

We’re pleased to inform you that your manuscript has been judged scientifically suitable for publication and will be formally accepted for publication once it meets all outstanding technical requirements.

Kind regards,

Peter F. Biehl, PhD

Academic Editor

PLOS ONE
---

## [Editor Report · Acceptance letter]

18 Jan 2024

PONE-D-23-20645R1 

PLOS ONE

Dear Dr. Fischer, 

I'm pleased to inform you that your manuscript has been deemed suitable for publication in PLOS ONE. Congratulations! Your manuscript is now being handed over to our production team.

Kind regards, 

on behalf of

Dr. Peter F. Biehl 

Academic Editor

PLOS ONE